# Sensitive period for rescuing parvalbumin interneurons connectivity and social behavior deficits caused by *TSC1* loss

Clara A. Amegandjin[1,2,5], Mayukh Choudhury[1,2,5], Vidya Jadhav[1,2], Josianne Nunes Carriço [2], Ariane Quintal[1], Martin Berryer[1,2], Marina Snapyan[3,4], Bidisha Chattopadhyaya[2], Armen Saghatelyan [3,4] & Graziella Di Cristo [1,2✉]

The Mechanistic Target Of Rapamycin Complex 1 (mTORC1) pathway controls several aspects of neuronal development. Mutations in regulators of mTORC1, such as Tsc1 and Tsc2, lead to neurodevelopmental disorders associated with autism, intellectual disabilities and epilepsy. The correct development of inhibitory interneurons is crucial for functional circuits. In particular, the axonal arborisation and synapse density of parvalbumin (PV)-positive GABAergic interneurons change in the postnatal brain. How and whether mTORC1 signaling affects PV cell development is unknown. Here, we show that Tsc1 haploinsufficiency causes a premature increase in terminal axonal branching and bouton density formed by mutant PV cells, followed by a loss of perisomatic innervation in adult mice. PV cell-restricted Tsc1 haploinsufficient and knockout mice show deficits in social behavior. Finally, we identify a sensitive period during the third postnatal week during which treatment with the mTOR inhibitor Rapamycin rescues deficits in both PV cell innervation and social behavior in adult conditional haploinsufficient mice. Our findings reveal a role of mTORC1 signaling in the regulation of the developmental time course and maintenance of cortical PV cell connectivity and support a mechanistic basis for the targeted rescue of autism-related behaviors in disorders associated with deregulated mTORC1 signaling.

[1] Neurosciences Department, Université de Montréal, Pavillon Paul-G.-Desmarais, Montréal, QC, Canada. [2] Centre de Recherche, CHU Sainte-Justine (CHUSJ), Montréal, QC, Canada. [3] CERVO Brain Research Center, Québec, QC, Canada. [4] Department of Psychiatry and Neuroscience, Université Laval, Québec, QC, Canada. [5] These authors contributed equally: Clara A. Amegandjin, Mayukh Choudhury. ✉email: graziella.di.cristo@umontreal.ca

The mechanistic target Of Rapamycin Complex 1 (mTORC1) acts as a central hub integrating internal and external stimuli to regulate many critical cellular processes, including cell growth and metabolism, protein synthesis, and autophagy[1]. mTORC1 signaling has also emerged as an important regulator of brain development and plasticity. Deregulation of mTORC1 signaling network is at the basis of several genetic neurodevelopmental disorders, which share common clinical features, such as epilepsy, autism, and other comorbidities[2,3]. In particular, mutations in the mTORC1 negative regulators *TSC1* or *TSC2* cause tuberous sclerosis complex (TSC), an autosomal dominant disease associated with high occurrence of epilepsy, intellectual disabilities, and autistic traits[4]. Extensive studies on TSC mutations have set the paradigm for monogenic "mTOR-pathies", to understand how mTOR dysregulation affects different processes of brain development[3] and how these may ultimately lead to cognitive and neurological deficits.

The theory of an increased excitation/inhibition (E/I) ratio as an underlying cause of network hyper-excitability and reduced signal-to-noise in the cortex was initially proposed by Rubenstein and Merzenich as a framework for understanding the pathophysiology of autism[5]. Over the past 15 years, numerous studies have provided evidence that alterations in E/I balance may be involved in many mouse models of monogenetic autism, however the nature of the underlying mechanisms are heterogeneous thus highlighting that it is critical to understand what sort of circuit alterations are caused by specific genetic mutations[6]. While numerous studies have focussed on the effects of *Tsc1/2* deletion, and mTOR dysregulation, on cortical and hippocampal excitatory cells[7–10], only few studies have addressed whether and how *Tsc1/2* deletion affects cortical GABAergic circuit development[11–15]. In particular, whether it plays different roles in specific GABAergic populations is not known.

The neocortex is comprised of a diverse group of inhibitory neurons, which differ in morphology, intrinsic physiological properties, and connectivity[16]. Among them, parvalbumin (PV) expressing cells, which represent the largest class of cortical interneurons, specifically target the soma and proximal dendrites of pyramidal cells, and have been implicated in synchronizing the firing of neuronal populations to generate gamma oscillation[17–19], which in turn allows the cortex to perform precise computational tasks underlying perception, selective attention, working memory, and cognitive flexibility in humans and rodents[20–23]. The development of PV cell circuit connectivity is a prolonged process, terminating around the end of adolescence in rodents and primates[24–28]. PV cells dysfunction has been found in several mouse models of autism[29–33]. Conversely, stimulating PV cells has been shown to be sufficient to ameliorate social behavior[29,34,35]. Since mutations in Tsc1 give rise to autistic traits, we questioned whether and how *Tsc1* deletion selectively in PV cells affects their connectivity, and whether and to what extent these alterations in cortical PV cell circuits might be contributing to changes in social behavior downstream of altered mTOR signaling.

Here, we used a combination of single-cell genetics in cortical organotypic cultures, conditional mutant mice, and high-resolution imaging to investigate the effects of TSC-mTORC1 pathway on the development of PV cell connectivity. We found that mutant PV cells (both heterozygous and homozygous) showed a premature increase of their axonal arbor complexity and bouton density in the first three postnatal weeks, followed by a striking loss of connectivity by adulthood. The effect of Tsc1 haploinsufficiency or deletion on PV cell connectivity was cell-autonomous. Further, conditional mutant mice showed social behavior deficits. Strikingly, both PV cell connectivity and social behavior in adult mice were rescued by a short treatment with the mTORC1 inhibitor rapamycin during the third postnatal week, suggesting that inhibiting the premature maturation of PV cell innervations was sufficient to ameliorate the long-term neurological outcomes of the mutation.

## Results

**TSC1 haploinsufficiency in postnatal PV cells reduced PV cell connectivity and altered social behavior in adulthood.** The maturation of PV cell innervation is a prolonged process that plateaus at the end of the first postnatal month in mouse cortex[24]. To investigate whether mTORC1 activation plays a role in this process, we first analyzed the time course of pS6 expression, one of the direct downstream effectors of mTORC1, in PV cells identified by PV immunolabeling (Fig. 1a). We found that both the proportion of PV cells expressing pS6 (Fig. 1a, b) and the mean intensity of pS6 signal (Fig. 1c) significantly increased between the third and fourth postnatal weeks in the somatosensory cortex. After the fourth postnatal week, the proportion of PV cells expressing pS6 remained stable (P26: 75 ± 7%, P35: 70 ± 7%; P150: 78 ± 3%, one-way Anova, $p > 0.1$; $n = 3$ for each age group). To investigate whether the increase of pS6 expression levels was a generalized phenomenon during this developmental window, we quantified pS6 levels in NeuN+ neurons that represent for the most part pyramidal cells in the cortex (Fig. 1d). We found no significant difference in the number of NeuN+ cells expressing pS6 between P18 and P26 (Fig. 1e).

Since this developmental time window coincides with the peak of the formation of rich and complex perisomatic GABAergic synapse innervation[24,26], a process that is highly modulated by neuronal activity and sensory experience[24,27], we asked whether and how dysregulation of the TSC-mTOR pathway affects the development of PV cell connectivity.

To answer this question, we used a transgenic mouse carrying a conditional allele of *Tsc1*[36], which allows cell-specific developmental stage restricted manipulation of *Tsc1*, crossed to the mouse line with the Cre allele under the control of the PV promoter ($PV$-$Cre^{+/-}$). This cross generated PV-cell restricted homozygous ($PV$-$Cre^{+/-}$; $Tsc1^{flox/flox}$) and heterozygous ($PV$-$Cre^{+/-}$; $Tsc1^{flox/+}$) mice and their control $PV$-$Cre^{-/-}$ littermates ($PV$-$Cre^{-/-}$; $Tsc1^{flox/flox}$ and $PV$-$Cre^{-/-}$; $Tsc1^{flox/+}$ mice, referred to hereafter as $Tsc1^{Ctrl}$).

To confirm the time course and specificity of Cre expression in *PV-Cre* mice, we used the $RCE^{GFP}$ reporter mouse. We observed that about 35% (35 ± 8.11%; $n = 4$ mice) of all PV cells expressed GFP by P14 in the somatosensory cortex, which rose to around 75% (80.43 ± 6.78%; $n = 8$ mice) in P20 mice and to 90% (94.72 ± 2.22%; $n = 8$) in P70 mice, which is consistent with previous findings that PV expression peaks by the third postnatal week. In addition, we confirmed the specificity of Cre expression, since virtually all GFP+ cells expressed PV at all the analyzed ages (98.60 ± 0.50% at P14, 98.21 ± 1.78% at P20; 98.02 ± 0.49% at P70). To control for the efficiency of *Tsc1* deletion in PV cells, we analyzed both pS6 expression and soma size, since both increase following mTORC1 hyperactivity[7,37,38]. At P45, we observed a higher proportion of PV cells co-localized with pS6 (Supplementary Fig. 1a, b) and a 2.5-fold increase in pS6 intensity in PV cell somata from *PV-Cre; Tsc1^{flox/flox}* mice (Supplementary Fig. 1c) as compared to control mice, while the soma size was significantly increased in both mutant genotypes (Supplementary Fig. 1e). Here, we used pS6 as a proxy measure of mTORC1 activation, however mTORC1 downstream signaling including S6 kinase can be influenced by multiple pathways, which may explain why we could not detect increased pS6 in the PV cell somata of *PV-Cre; Tsc1^{flox/+}* mice. Alternatively, changes in pS6 might be more localized to specific cellular subdomains in the conditional heterozygous mouse.

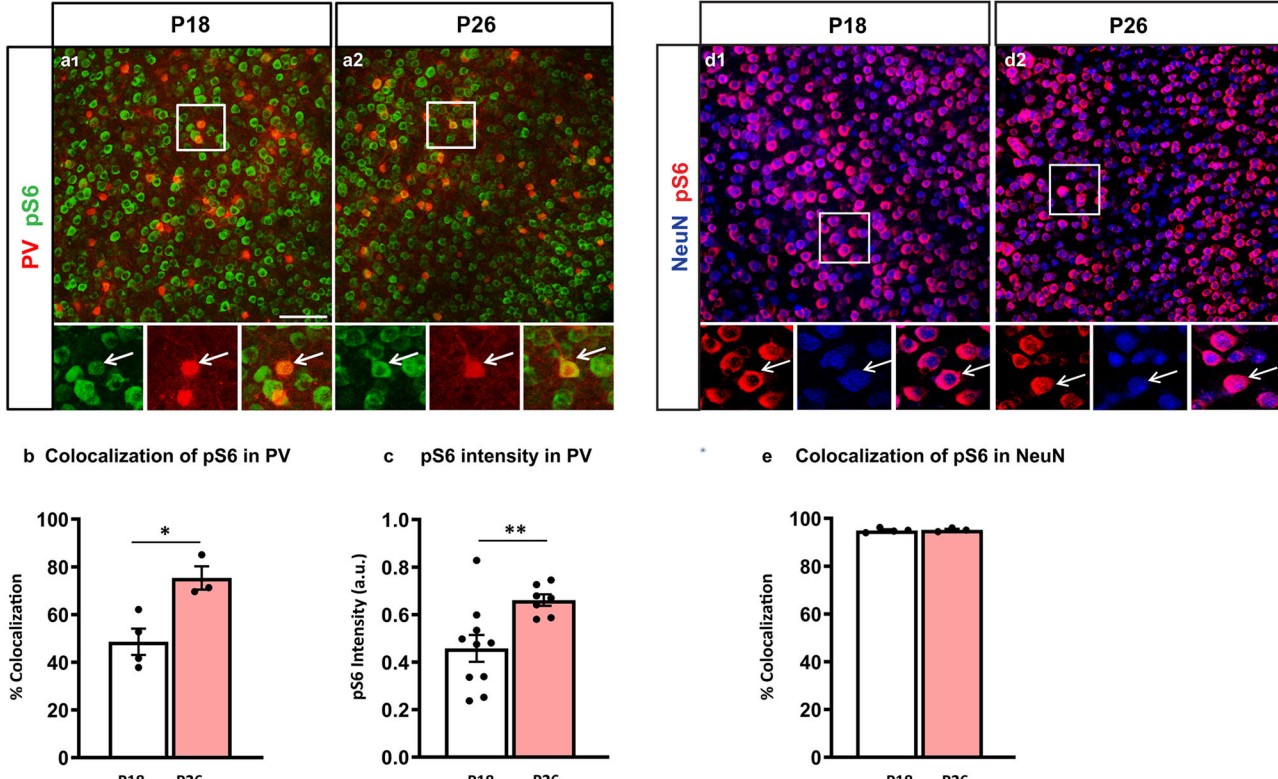

**Fig. 1 pS6 expression levels increase specifically in PV cells between the second and fourth postnatal weeks. a** Coronal sections of mouse somatosensory cortex immunostained for pS6 (green) and PV (red) at P18 (**a1**) and P26 (**a2**). **b** Number of PV cells expressing detectable levels of pS6 increases during the second to fourth postnatal week (Welch's $t$-test, $*p = 0.0152$). Number of mice; P18, $n = 4$; P26, $n = 3$. **c** Mean pS6 intensity in individual PV cells is significantly higher at P26 than at P18 (Welch's $t$-test, $**p = 0.0061$). Number of mice; P18, $n = 10$; P26 $n = 7$. **d** Coronal sections of mouse somatosensory cortex immunostained for pS6 (red) and NeuN (blue) at P18 (**d1**) and P26 (**d2**). **e** Percentage of colocalization of pS6 and NeuN is not significantly different between the two developmental ages (Welch's $t$-test, $p = 0.7663$). Number of mice; P18, $n = 4$; P26, $n = 3$. Scale bars in **a1-a2**, **d1-d2**, 75 μm. Data represent mean ± SEM. Source data are provided as a Source Data file.

To determine if in vivo postnatal loss of *Tsc1* leads to defects in PV cell connectivity, we quantified perisomatic PV synapse density by (a) immunostaining cortical slices with presynaptic (PV) and postsynaptic (gephyrin) markers and (b) by EM analysis of PV+ axons and synapses. We found that the density of perisomatic PV+/gephrin+ punctas was significantly and comparably decreased in mice heterozygous and homozygous for the conditional *Tsc1* allele (Fig. 2a–d). In addition, the density of PV+ terminals and the length of PV+ synapses were significantly reduced in *PV-Cre;Tsc1flox/+* mice (Fig. 2e–l) suggesting that PV-cell restricted, postnatal *Tsc1* haploinsufficiency leads to PV cell hypo-connectivity in adult mice. It has been reported that *Tsc1* deletion in cortical GABAergic neurons[14] or Purkinje cells[38] leads to neuronal loss in the targeted population. In our hands, we did not observe any difference in cortical PV cell density in our conditional mutant mice vs control littermates (PV/NeuN; *Tsc1*Ctrl mice: 11.5 ± 1.2%, $n = 6$ mice; *PV-Cre;Tsc1flox/+*: 10.4 ± 1.1%, $n = 4$ mice, *PV-Cre;Tsc1flox/flox*: 11.6 ± 0.4%, $n = 5$ mice), emphasizing that the observed PV cell hypo-connectivity in adult mutant mice was not due to PV cell loss.

To investigate whether *PV-Cre; Tsc1flox* mutants might show abnormal behaviors resembling autism spectrum disorders (ASDs), we evaluated social interaction using the three-chamber assay of social approach and preference for social novelty. In contrast to what observed in wild-type littermates, both heterozygous and homozygous mutant mice showed no significant differences in the time spent interacting with a mouse vs an object (Fig. 2o) or with a novel vs a familiar mouse

(Fig. 2p). This phenotype was not due to major motor problems or increased anxiety, since there were no differences in locomotor activity, as tested in the open field (Fig. 2m), and no increased anxiety in the elevated plus maze paradigm (Fig. 2n), between the wild-type and mutant mice. In fact, *PV-Cre;Tsc1flox/flox* but not *PV-Cre;Tsc1flox/+* mice exhibit less anxiety-like behavior, since they spent significantly more time in the open arms (Fig. 2n).

In summary, *Tsc1* deletion in postnatal PV cells leads to alterations in both PV cell connectivity and in social behavior. PV cell hypo-connectivity in adult mutant mice could be directly caused by Tsc1-mTORC1 signaling dysregulation in PV cells or induced as a consequence of homeostatic feedback mechanisms that influence neural circuit development. We then used both in vitro and in vivo approaches to determine the cell-autonomous and network phenotypes resulting from the genetic deletion of *Tsc1* in cortical PV cells.

**mTORC1 hyper-activation in single PV cells induced a premature increase in bouton density and axon branching, subsequently followed by excessive bouton pruning.** To explore the cell autonomous effects of *Tsc1* deletion or haploinsufficiency during specific developmental phases of PV cell connectivity, we used single cell genetic manipulation in cortical organotypic cultures (Supplementary Fig. 2a). To reduce *Tsc1* expression in single PV cells and simultaneously labeling their axons and synapses, we used a previously characterized promoter region $P_{G67}$[24] to express either Cre recombinase together with GFP ($P_{G67}$-GFP/Cre) or GFP alone (control) in single PV cells in

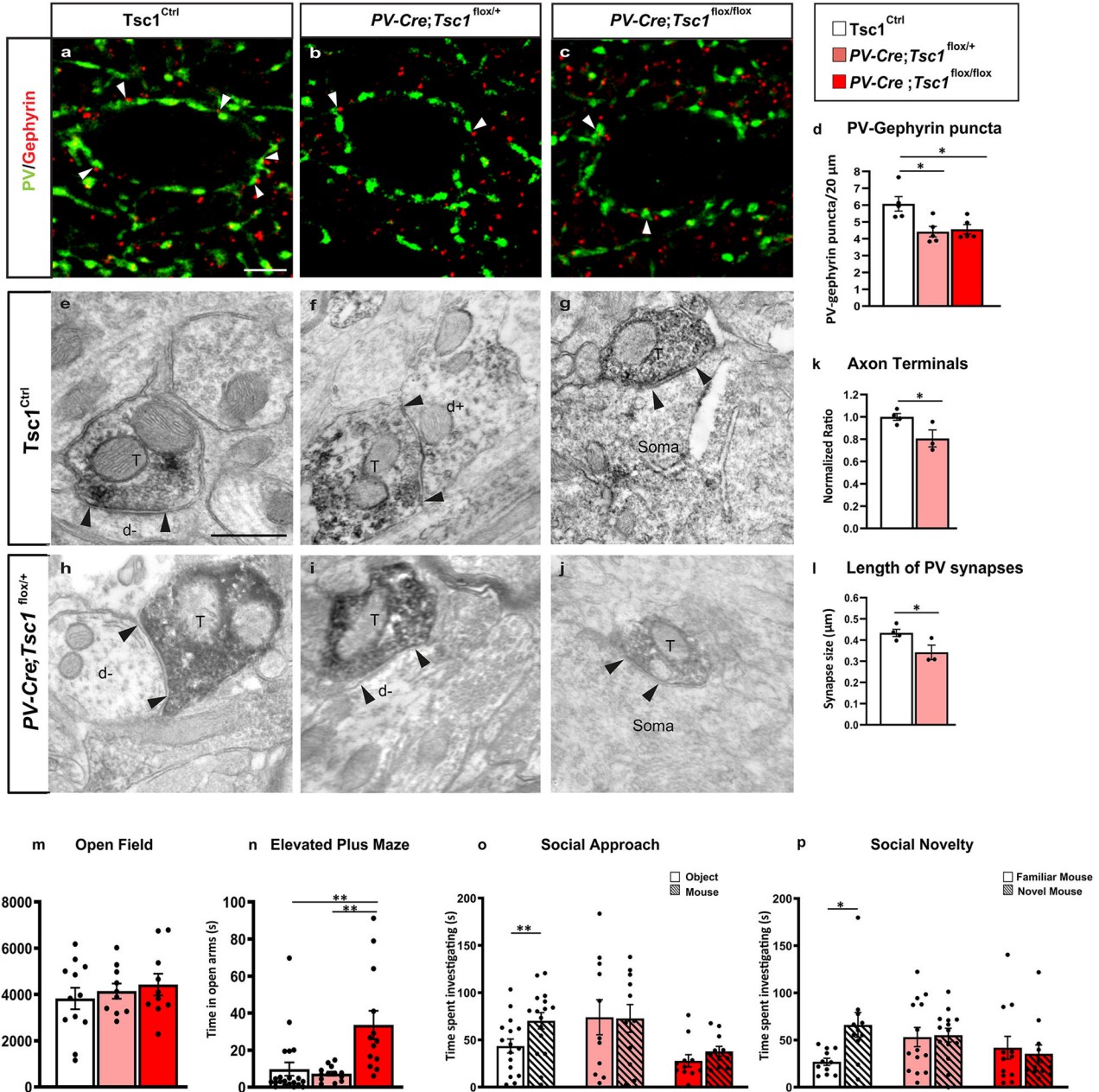

**Fig. 2 Tsc1 knockout in PV cells causes PV cell hypo-connectivity and social behavioral deficits in young adult mice. a–c** Somatosensory cortex coronal sections immunostained for PV (green) and gephyrin (red) from P60 *Tsc1*<sup>Ctrl</sup> (**a**), *PV-Cre;Tsc1*<sup>flox/+</sup> (**b**), and *PV-Cre;Tsc1*<sup>flox/flox</sup> mice (**c**). White arrowheads indicate PV-gephyrin colocalized boutons. **d** PV/gephyrin colocalized puncta (one-way ANOVA, *$p = 0.0096$; Tukey's multiple comparisons test: *Tsc1*<sup>Ctrl</sup> vs *PV-Cre;Tsc1*<sup>flox/+</sup> *$p = 0.0140$; *Tsc1*<sup>Ctrl</sup> vs *PV-Cre;Tsc1*<sup>flox/flox</sup> *$p = 0.0236$), $n = 5$ mice for all genotypes. Scale bar: 10 µm. **e-j** PV-immunolabeled axon terminals in somatosensory cortex of *Tsc1*<sup>Ctrl</sup> (**e, f, g**) and *PV-Cre;Tsc1*<sup>flox/+</sup> (**h, i, j**) mice. **e–g** PV + axon terminals (T) make symmetric synaptic contact (flanked by arrowheads) with an unlabeled dendritic shaft (d−). **f** Rare symmetric synapse between PV + axon terminal and PV + dendrite (d+). **g, j** Synaptic contact between a PV + axon terminal (T) and an unlabeled cell soma. **k** Quantification of PV + axon terminals in *PV-Cre;Tsc1*<sup>flox/+</sup> mice (Unpaired *t*-test, *$p = 0.0453$). **l** *PV-Cre; Tsc1*<sup>flox/+</sup> mice have shorter synapses than *Tsc1*<sup>Ctrl</sup> (Unpaired *t* test, *$p = 0.0464$). Number of mice: *Tsc1*<sup>Ctrl</sup> $n = 4$; *PV-Cre;Tsc1*<sup>flox/+</sup> $n = 3$. Scale bars: **e–j**, 500 nm. **m** Open field test: distance traveled during exploratory activity in an open field arena (one-way ANOVA, $p > 0.05$; Tukey's multiple comparisons test: *Tsc1*<sup>Ctrl</sup> vs *PV-Cre;Tsc1*<sup>flox/+</sup> $p = 0.8537$; *Tsc1*<sup>Ctrl</sup> vs *PV-Cre;Tsc1*<sup>flox/flox</sup> $p = 0.5794$; *PV-Cre;Tsc1*<sup>flox/+</sup> vs *PV-Cre;Tsc1*<sup>flox/flox</sup> $p = 0.8956$). Number of mice: *Tsc1*<sup>Ctrl</sup> $n = 12$; *PV-Cre;Tsc1*<sup>flox/+</sup> $n = 10$; *PV-Cre;Tsc1*<sup>flox/flox</sup> $n = 10$. **n** Elevated plus maze: Quantification of time spent in the open arms of elevated plus maze arena (one-way ANOVA, ***$p = 0.0006$; Tukey's multiple comparisons test: *Tsc1*<sup>Ctrl</sup> vs *PV-Cre;Tsc1*<sup>flox/+</sup> $p = 0.9205$; *Tsc1*<sup>Ctrl</sup> vs *PV-Cre;Tsc1*<sup>flox/flox</sup> **$p = 0.0017$; *PV-Cre; Tsc1*<sup>flox/+</sup> vs *PV-Cre;Tsc1*<sup>flox/flox</sup> **$p = 0.0018$; Number of mice: *Tsc1*<sup>Ctrl</sup> $n = 21$; *PV-Cre;Tsc1*<sup>flox/+</sup> $n = 13$; *PV-Cre;Tsc1*<sup>flox/flox</sup> $n = 13$. **o, p** Unlike *Tsc1*<sup>Ctrl</sup> mice, both *PV-Cre;Tsc1*<sup>flox/+</sup> and *PV-Cre;Tsc1*<sup>flox/flox</sup> mice failed to show preference for a mouse vs an object (**o**) (two-way ANOVA, $F_{genotype}$ (2, 35) = 3.968 $p = 0.0280$, $F_{time}$ (1, 35) = 4.593 $p = 0.0391$, $F_{genotype*time}$ (2, 35) = 2.376 $p = 0.1077$; **$p = 0.0089$, Sidak's multiple comparisons test) or for a novel mouse vs a familiar one (**p**) (two-way ANOVA, $F_{genotype}$ (2, 34) = 1.108 $p = 0.3418$, $F_{time}$ (1, 34) = 2.640 $p = 0.1134$, $F_{genotype*time}$ (2, 34) = 3.615 $p = 0.0377$; *$p = 0.0147$, Sidak's multiple comparisons test); Number of mice: **o** *Tsc1*<sup>Ctrl</sup> $n = 16$; *PV-Cre;Tsc1*<sup>flox/+</sup> $n = 11$; *PV-Cre;Tsc1*<sup>flox/flox</sup> $n = 11$. **p** *Tsc1*<sup>Ctrl</sup> $n = 11$; *PV-Cre;Tsc1*<sup>flox/+</sup> $n = 14$; *PV-Cre;Tsc1*<sup>flox/flox</sup> $n = 12$. Data represent mean ± SEM. Source data are provided as a Source Data file.

cortical organotypic cultures from $Tsc1^{flox/flox}$ and $Tsc1^{flox/+}$ mice. This approach allowed us to generate specifically $Tsc1^{-/-}$ and $Tsc1^{+/-}$ PV cells in an otherwise wild-type background. Deletions of either one or both $Tsc1$ alleles significantly increased pS6 expression levels in the transfected sparse PV cells (Supplementary Fig. 2e), while cell soma size was significantly increased only in $Tsc1^{-/-}$ PV cells (Supplementary Fig. 2f).

We have previously shown that the basic features of the mature perisomatic innervation formed by PV cells onto pyramidal cells are recapitulated in cortical organotypic cultures[24,39]. PV innervation starts out with simple axons, which develop into complex, highly branched arbors in the subsequent 4 weeks with a time course similar to that observed in vivo[24]. In particular, PV cell axonal arborization and bouton density increase significantly between EP18 (P5 + 13 days in vitro = equivalent postnatal day 18) and EP24. To investigate the effect of premature mTORC1 activation on PV cell synapse innervation, we biolistically transfected PV cells at EP10 and analyzed them at EP18 (Supplementary Fig. 2a). Following $Tsc1$ deletion, we quantified two aspects of individual PV cell connectivity—(1) the extent of perisomatic innervation around single targeted somata (terminal branching and perisomatic bouton density) and (2) the fraction of potentially innervated somata within the basket cell arbor (percentage of innervation or the innervation field). We have previously shown that the vast majority of GFP-labeled boutons in our experimental conditions most likely represent presynaptic terminals[24,25,40]. We found that both $Tsc1^{-/-}$ and $Tsc1^{+/-}$ PV cells formed premature perisomatic innervations, characterized by increased bouton density (Fig. 3a, b, e and Supplementary Fig. 3a) and terminal axonal branching around NeuN+ contacted somata (Fig. 3f and Supplementary Fig. 3b), and increased percentage of contacted target cells (Fig. 3h). To determine whether the effects of $Tsc1$ deletion are due to mTORC1 hyperactivation, we treated cortical organotypic cultures with the mTORC1 inhibitor Rapamycin from EP10–18 (90 ng/ml, Supplementary Fig. 4) and found that Rapamycin treatment reversed the increase in bouton density in $Tsc1^{-/-}$ PV cells (Supplementary Fig. 4e) as well as terminal branching (Supplementary Fig. 4f). All together, these data suggest that mTORC1 hyper activation leads to the premature formation of PV cell innervations in a cell-autonomous manner.

Next, we asked whether the premature development of PV cell innervation was long lasting. As described above, PV cells were transfected at EP10 and then analyzed either at EP24 (Supplementary Fig. 5, during the peak of the proliferation of PV cell innervation) or at EP34 (Fig. 3c, d, when PV cell innervation has matured and is stable). At EP24, perisomatic innervation formed by $Tsc1^{-/-}$ PV cells were similar to those formed by age-matched wild-type cells (Supplementary Fig. 5). However, at EP34, $Tsc1^{-/-}$ PV cells showed significantly poorer innervations than age-matched $Tsc1^{+/+}$ PV cells (Fig. 3e, g, h). All together, these data show that dysregulated TSC-mTORC1 signaling in individual PV cells alters the development of their innervations, inducing first a premature increase in axonal branching and bouton density followed by excessive pruning (Fig. 3i), in a cell-autonomous fashion.

Next, we sought to investigate whether the phenotypic switch of PV cell connectivity caused by $Tsc1$ deletion occurs in vivo. Since Cre expression in *PVCre* mice starts at around P10 and only peaks towards the fourth postnatal week, we reasoned that the time course of $Tsc1$ allele recombination, and its effects on PV cell innervation, could be highly variable between P18 and P24. To overcome this issue, we generated Tg($Nkx2.1$-$Cre$);$Tsc1^{flox}$ and control littermates. NKX2.1 is a transcription factor expressed at E10.5 by GABAergic cell precursors in the medial ganglionic eminence (MGE), which gives rise to cortical PV- and somatostatin-expressing (SST) GABAergic cells[41]. At P18, pS6

levels and soma size were significantly increased in PV cells from Tg($Nkx2.1$-$Cre$);$Tsc1^{flox/flox}$ mouse somatosensory cortex (Supplementary Fig. 6a–c, e). By P45, PV cells showed a four and twofold increase in pS6 intensity in Tg($Nkx2.1$-$Cre$);$Tsc1^{flox/flox}$ and Tg($Nkx2.1$-$Cre$);$Tsc1^{flox/+}$ mice compared to control mice, respectively (Supplementary Fig. 6d). PV cell somata were larger in Tg($Nkx2.1$-$Cre$);$Tsc1^{flox/+}$ compared to $Tsc1^{Ctrl}$ mice, even if not as large as those in Tg($Nkx2.1$-$Cre$);$Tsc1^{flox/flox}$ mice (Supplementary Fig. 6f), suggesting that deletion of one $Tsc1$ allele may have slow, cumulative effects in vivo, consistent to what is previously reported in Purkinje cell-specific $Tsc1$ mutant mice[38]. A previous study showed that conditional $Tsc1$ deletion in GABAergic progenitor cells using Dlx5/6-Cre mice leads to reduced cortical GABAergic cell density[14]. Conversely, we found no difference in PV cell density between the mutant mice and control littermates at P18 (PV/NeuN; Ctrl mice: 8.9 ± 0.7%; $n = 3$ mice; $Nkx2.1$-$Cre$;$Tsc1^{flox/+}$: 8.1 ± 0.3%; $n = 3$ mice, $Nkx2.1$-$Cre$; $Tsc1^{flox/flox}$: 7.9 ± 0.3%; $n = 4$ mice). This difference suggests that conditional deletion of $Tsc1$ at the time of cell cycle exit (Nkx2.1-Cre) has a different impact than removal on the mantle zone (Dlx5/6-Cre) on GABAergic neuron survival.

In order to analyze the PV cell axonal morphology at high resolution we turned to organotypic cultures from Tg($Nkx2.1$-$Cre$); $Tsc1^{flox/flox}$, Tg($Nkx2.1$-$Cre$);$Tsc1^{flox/+}$, and $Tsc1^{Ctrl}$ mice transfected with $P_{G67}$-GFP at different developmental stages (Figs. 4, 5). At EP18, before the peak of PV cell synapse proliferation, similar to what we observed with the single cell $Tsc1$ deletion, we found that PV cells from both Tg($Nkx2.1$-$Cre$); $Tsc1^{flox/flox}$ and Tg($Nkx2.1$-$Cre$);$Tsc1^{flox/+}$ mice formed more complex perisomatic innervations, characterized by increased perisomatic bouton density (Fig. 4a–d) and terminal branching (Fig. 4e) compared to age-matched control PV cells in cultures transfected from $Tsc1^{Ctrl}$ mice.

Conversely, at EP34, in a period when PV axonal arbor maturation has reached stability, PV neurons from both genotypes (Tg($Nkx2.1$-$Cre$);$Tsc1^{flox/flox}$, Tg($Nkx2.1$-$Cre$);$Tsc1^{flox/+}$) showed significantly reduced perisomatic bouton density (Fig. 5a–d), terminal branching (Fig. 5e), and innervated a smaller percentage of pyramidal neurons (Fig. 5f). These results are consistent with a recent study showing that pyramidal cell synaptic inhibition is reduced in the hippocampus of adult $Nkx2.1$-$Cre$;$Tsc1^{flox/+}$ mice[15]. Overall, these results confirm that embryonic deletion of $Tsc1$ has opposite effects on PV perisomatic synapse formation and maintenance, initially accelerating the formation of PV synaptic innervation and subsequently impairing perisomatic synapses at the maturation phase. Further, deletion of a single $Tsc1$ allele in PV cells is sufficient to alter its connectivity both at the single cell and network levels.

Furthermore, behavioral analysis showed that both heterozygous and homozygous Tg($Nkx2.1$-$Cre$);$Tsc1^{flox}$ mice phenocopied the deficits in social behavior (social approach and social novelty preference) that we found in $PV$-$Cre$;$Tsc1^{flox}$ mice (Supplementary Fig. 7), while Tg($Nkx2.1$-$Cre$);$Tsc1^{flox/flox}$ and Tg($PV$-$Cre$);$Tsc1^{flox/flox}$ showed an opposite phenotype in the elevated plus maze, which may due to the different GABAergic circuits affected in the two mouse lines (Nkx2.1 is expressed by somatostartin-positive neurons as well) or/and by the different timing of $Tsc1$ deletion (embryonal vs postnatal).

**Tsc1 deletion in GABAergic cells causes transient autophagy dysfunction in adolescent mice.** A recent study revealed that, in neurons, Tsc2/1 loss altered the process of autophagy by AMP-activated kinase (AMPK)-dependent mechanisms[42]. To investigate whether autophagy was affected by conditional $Tsc1$ deletion in GABAergic interneurons, we analyzed by western blot the

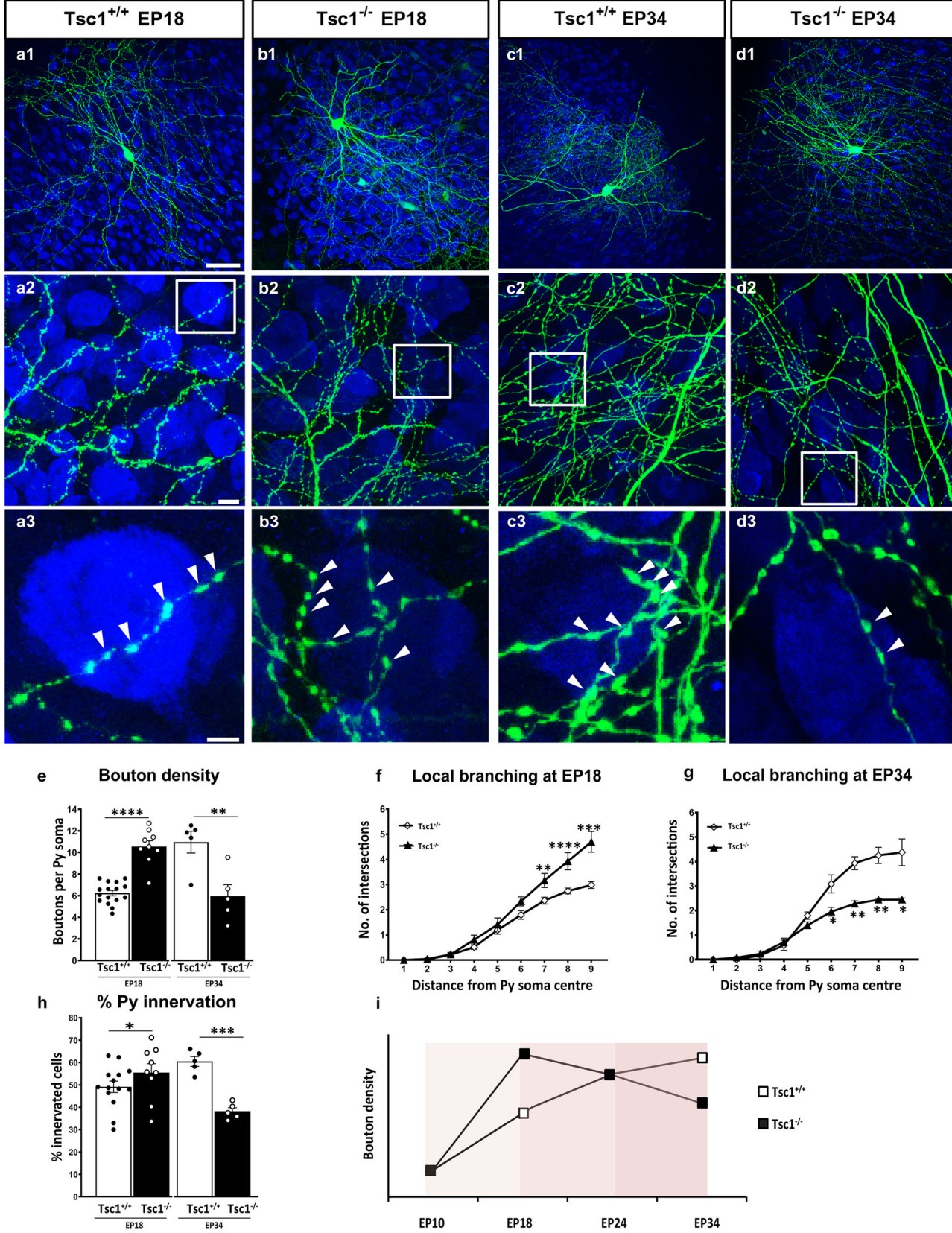

expression levels of the autophagosomal lipidated microtubule-associated protein 1 light chain 2 (LC3-II), of the autophagy substrate p62/sequestosome 1 and of the autophagy-initiating kinase Unc-51-like-kinase (ULK1) in Tg(*Nkx2.1Cre*);*Tsc1* $^{flox/flox}$ mice compared to that of their control littermates in young and adult mice. Since GABAergic neurons constitute a minority of cortical cells, we extracted proteins from the olfactory bulb where GABAergic cells are highly enriched, to increase the likelihood of detecting small changes in protein expression levels. We found significantly increased levels of LC3-II, but not of p62, in young

**Fig. 3 *Tsc1* knockout in single PV neurons causes a premature increase in axonal terminal branching and bouton density followed by excessive pruning.** **a1** EP18 *Tsc1*[+/+] PV cell showing characteristic branching (**a2**) and boutons (**a3**, arrowheads) on the postsynaptic somata identified by NeuN immunostaining (blue). **b** *Tsc1*[−/−] PV cells lacking both alleles (**b1–b3**) of *Tsc1* show significant increase in bouton density at EP18. **c** Control EP34 *Tsc1*[+/+] PV cell. **d1–d3** EP34 *Tsc1*[−/−] PV cell showing significantly decreased axonal branching (**c2** vs **d2**) and perisomatic boutons (**c3** vs **d3**). **e** *Tsc1*[−/−] PV cells show an increase in bouton density at EP18 (Welch's t-test, ****p < 0.0001) followed by a decrease at EP34 (Welch's t-test, **p = 0.0091). Number of PV cells: At EP18; n = 16 for *Tsc1*[+/+], n = 9 for *Tsc1*[−/−]. At EP34: n = 5 *Tsc1*[+/+], n = 5 *Tsc1*[−/−]. **f** *Tsc1*[−/−] PV cells show more developed branching than *Tsc1*[+/+] cells at EP18 (Welch's t-test, **p = 0.0017 (radius 7), ****p < 0.0001 (radius 8), ***p = 0.0001 (radius 9)). **g** At EP34 *Tsc1*[−/−] PV cells are characterized by simpler axonal branching compared to controls (Welch's t-test: *p = 0.0336 (radius 6), **p = 0.0016 (radius 7), **p = 0.0047 (radius 8), *p = 0.0232 (radius 9)). Number of PV cells: At EP18; n = 15 for *Tsc1*[+/+], n = 9 for *Tsc1*[−/−]. At EP34: n = 5 *Tsc1*[+/+], n = 5 *Tsc1*[−/−]. **h** Percentage of innervation is significantly increased in *Tsc1*[−/−] PV cells at EP18 and reduced at EP34 (Welch's t-test: EP18, *p = 0.0114; EP34, ****p < 0.0001). Number of PV cells: At EP18; n = 14 for *Tsc1*[+/+], n = 9 for *Tsc1*[−/−]. At EP34: n = 5 *Tsc1*[+/+], n = 5 *Tsc1*[−/−]. **i** Schematic representation of bouton density during the postnatal maturation of *Tsc1*[+/+] and *Tsc1*[−/−] PV cells Scale bars: a1–d1, 50 μm; a2–d2, 10 μm; a3–d3, 5 μm. Data represent mean ± SEM. Source data are provided as a Source Data file.

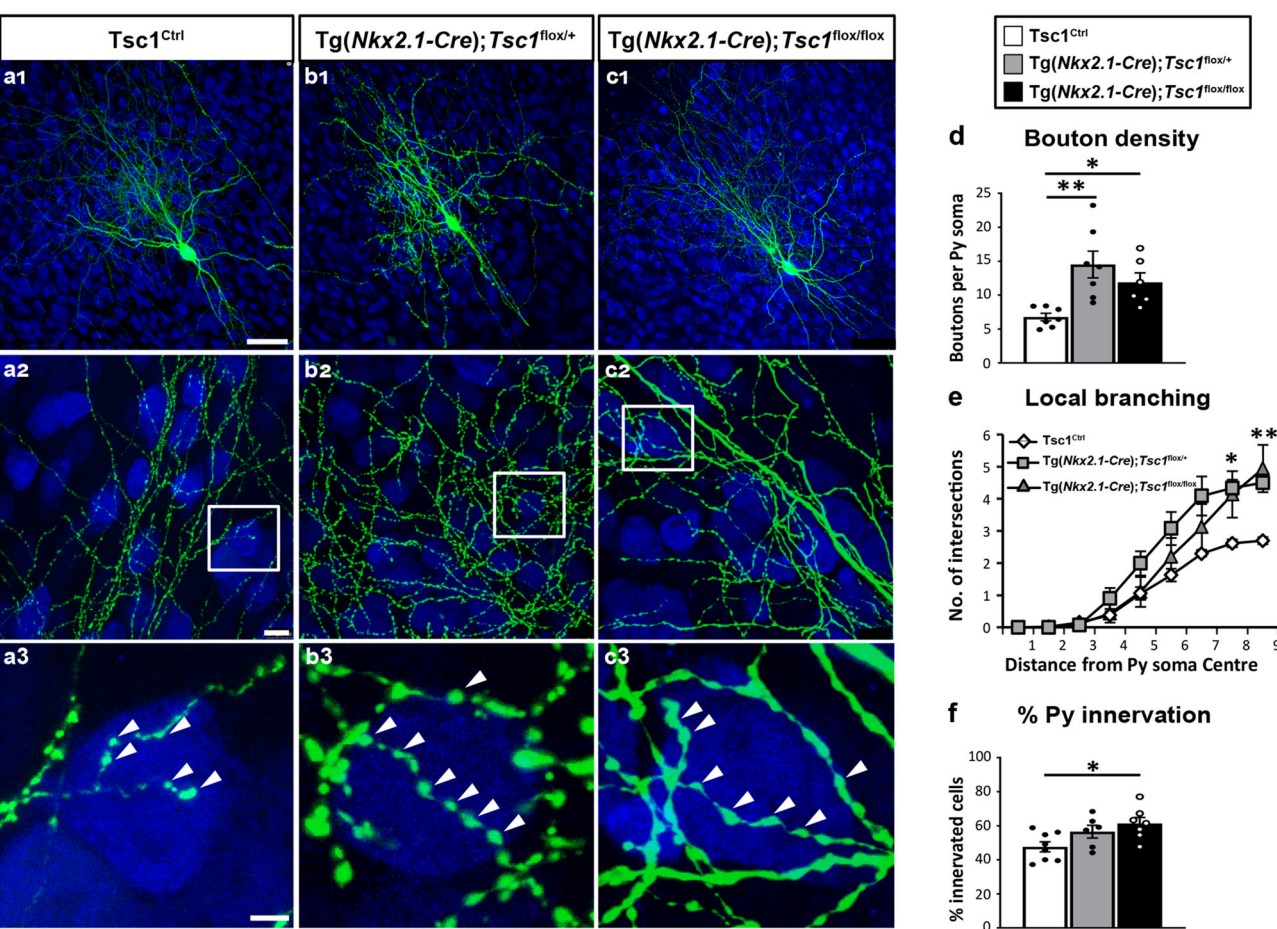

**Fig. 4 PV cells show prematurely rich perisomatic innervation in Tg(*Nkx2.1-Cre*);*Tsc1*[flox/flox] and Tg(*Nkx2.1-Cre*);*Tsc1*[flox/+] mice at EP18. a1** A PV cell (green) amongst NeuN immunostained neurons (in blue) in cortical organotypic culture from a *Tsc1*[Ctrl] mouse at EP18. **a2** PV cell from *Tsc1*[Ctrl] slice shows characteristic branching and multiple boutons (arrowheads) on the postsynaptic somata (**a3**). PV cells from Tg(*Nkx2.1-Cre*);*Tsc1*[flox/+] mice (**b1–b3**) and Tg(*Nkx2.1-Cre*);*Tsc1*[flox/flox] mice (**c1–c3**) show increased bouton density (**d**) (one-way ANOVA, **p = 0.0039; Holm–Sidak's multiple comparisons: *Tsc1*[Ctrl] vs Tg(*Nkx2.1-Cre*);*Tsc1*[flox/+] **p = 0.0023; *Tsc1*[Ctrl] vs Tg(*Nkx2.1-Cre*);*Tsc1*[flox/flox] *p = 0.0242). Number of mice: *Tsc1*[Ctrl] n = 7, Tg(*Nkx2.1-Cre*);*Tsc1*[flox/+] n = 7, Tg(*Nkx2.1-Cre*); *Tsc1*[flox/flox] n = 6. **e** Local branching (one-way ANOVA *p = 0.0113 (Radius 8), **p = 0.0096 (Radius 9); Holm–Sidak's multiple comparisons: (Radius 8) *Tsc1*[Ctrl] vs Tg(*Nkx2.1-Cre*); *Tsc1*[flox/+] *p = 0,0155; *Tsc1*[Ctrl] vs Tg(*Nkx2.1-Cre*); *Tsc1*[flox/flox] *p = 0.0425, Tg(*Nkx2.1-Cre*); *Tsc1*[flox/+] vs Tg(*Nkx2.1-Cre*); *Tsc1*[flox/flox] p = 0.8062; (Radius 9) *Tsc1*[Ctrl] vs Tg(*Nkx2.1-Cre*); *Tsc1*[flox/+] *p = 0.0317; *Tsc1*[Ctrl] vs Tg(*Nkx2.1-Cre*); *Tsc1*[flox/flox] *p = 0.0148, Tg(*Nkx2.1-Cre*); *Tsc1*[flox/+] vs Tg(*Nkx2.1-Cre*); *Tsc1*[flox/flox] p = 0.9738). Number of mice: *Tsc1*[Ctrl] n = 7, Tg(*Nkx2.1-Cre*);*Tsc1*[flox/+] n = 5, Tg(*Nkx2.1-Cre*); *Tsc1*[flox/flox] n = 6. **f** Percentage of innervation (one-way ANOVA, *p = 0.0254; Holm–Sidak's multiple comparisons: *Tsc1*[Ctrl] vs Tg(*Nkx2.1-Cre*); *Tsc1*[flox/+] p = 0,0823; *Tsc1*[Ctrl] vs Tg(*Nkx2.1-Cre*); *Tsc1*[flox/flox] *p = 0.0168). Number of PV cells: *Tsc1*[Ctrl] n = 8, Tg(*Nkx2.1-Cre*); *Tsc1*[flox/+] n = 6, Tg(*Nkx2.1-Cre*); *Tsc1*[flox/flox] n = 7. Scale bars: **a1–c1**, 20 μm; **a2–c2**, 10 μm, **a3–c3**, 3 μm. Data represent mean ± SEM. Source data are provided as a Source Data file.

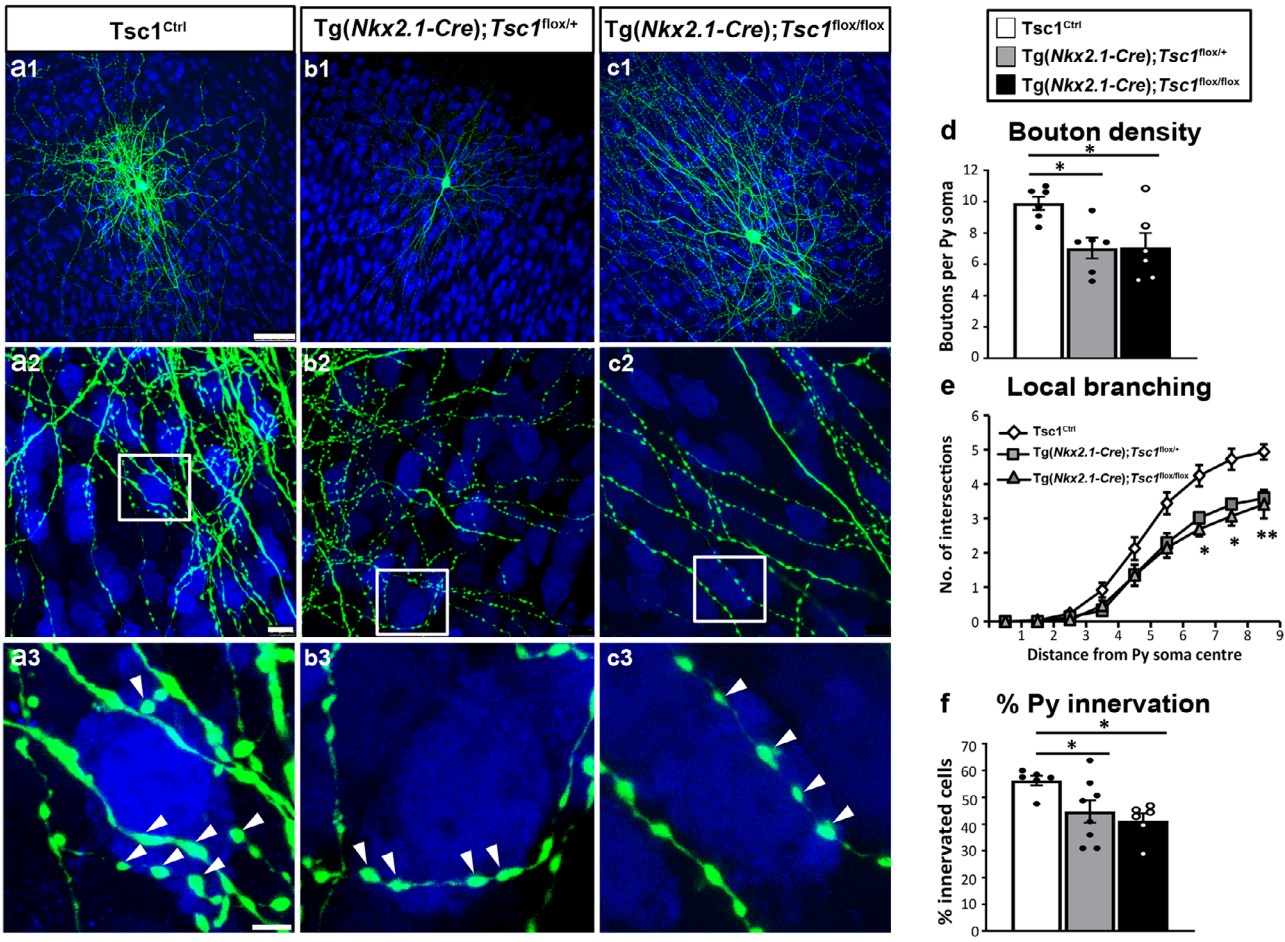

**Fig. 5 PV interneurons show significantly reduced perisomatic innervation in Tg(*Nkx2.1-Cre*);*Tsc1*flox/flox and Tg(*Nkx2.1-Cre*);*Tsc1*flox/+ mice at EP34. a** A PV cell (green) among NeuN immunostained neurons (blue) in cortical organotypic cultures from a *Tsc1*Ctrl mouse at EP34. **b**, **c** PV cells from Tg(*Nkx2.1-Cre*);*Tsc1*flox/+ mice (**b1**–**b3**) or Tg(*Nkx2.1-Cre*);*Tsc1*flox/flox mice (**c1**–**c3**) show decreased bouton density (**d**) (one-way ANOVA, *$p$ = 0.0157; Holm–Sidak's multiple comparisons: *Tsc1*Ctrl vs Tg(*Nkx2.1-Cre*);*Tsc1*flox/+ *$p$ = 0,0214; *Tsc1*Ctrl vs Tg(*Nkx2.1-Cre*); *Tsc1*flox/flox *$p$ = 0.0214). Local branching (**e**) (one-way ANOVA, **$p$ = 0.0040 (Radius 7), *$p$ = 0.0127 (Radius 8), **$p$ = 0.0011 (Radius 9); Holm–Sidak's multiple comparisons: (Radius 7) *Tsc1*Ctrl vs Tg(*Nkx2.1-Cre*);*Tsc1*flox/+ **$p$ = 0.0032; *Tsc1*Ctrl vs Tg(*Nkx2.1-Cre*); *Tsc1*flox/flox *$p$ = 0.0117; (Radius 8) *Tsc1*Ctrl vs Tg(*Nkx2.1-Cre*); *Tsc1*flox/+**$p$ = 0.0126; *Tsc1*Ctrl vs Tg(*Nkx2.1-Cre*); *Tsc1*flox/flox *$p$ = 0.0149; (Radius 9) *Tsc1*Ctrl vs Tg(*Nkx2.1-Cre*); *Tsc1*flox/+ ***$p$ = 0.0008; *Tsc1*Ctrl vs Tg(*Nkx2.1-Cre*);*Tsc1*flox/flox **$p$ = 0.0034). Number of mice: *Tsc1*Ctrl $n$ = 6, Tg(*Nkx2.1-Cre*);*Tsc1*flx/+ $n$ = 6, Tg(*Nkx2.1-Cre*); *Tsc1*flox/flox $n$ = 6. **f** Percentage of innervation is also significantly lower for PV cells from Tg(*Nkx2.1-Cre*);*Tsc1*flox/+ and Tg(*Nkx2.1-Cre*); *Tsc1*flox/flox mice (one-way ANOVA, *$p$ = 0.0212, Holm–Sidak's multiple comparisons: *Tsc1*Ctrl vs Tg(*Nkx2.1-Cre*);*Tsc1*flox/+ *$p$ = 0.0257; *Tsc1*Ctrl vs Tg(*Nkx2.1-Cre*);*Tsc1*flox/flox *$p$ = 0.0179). Number of PV cells: *Tsc1*Ctrl $n$ = 6, Tg(*Nkx2.1-Cre*);*Tsc1*flox/+ $n$ = 8, Tg(*Nkx2.1-Cre*);*Tsc1*flox/flox $n$ = 6. Arrowheads indicate boutons. Scale bars: **a1**–**c1**, 20 μm; **a2**–**c2**, 10 μm, **a3**–**c3**, 3 μm. Data represent mean ± SEM. Source data are provided as a Source Data file.

mutant mice compared to their littermates (Fig. 6a–d). We further found increased activation of AMPK, as indicated by the expression levels of phospho-AMPK (at T172) and a trend towards increased levels of pULK1 (at Ser555; Fig. 6e–g), consistent with what was previously reported[42]. Adult mice, on the other hand, did not show any significant alterations in any of these markers (Fig. 6h–l). To investigate whether GABAergic synapse density in the olfactory bulbs were affected by *Tsc1* deletion, we imaged the external plexiform layer (EPL), since this layer is highly enriched in PV cells[43]. In the EPL, PV-positive cells are typically axonless and their multipolar dendrites form dendro-dendritic synapses, which can be identified by immunolabeling for the vesicular GABA transporter (VGAT) and gephyrin[44]. Similar to what we observed for cortical PV+ Gephyrin+ density (Fig. 2a–d), we found a significant reduction in the density of vGAT+, gephyrin+ puncta in the EPL of adult mice compared to control littermates (Supplementary Fig. 8).

Overall, these data suggest that *Tsc1* loss in MGE-derived GABAergic cells leads to a dysregulation of autophagy and of AMPK activation during a critical developmental window, which overlaps temporally with the maturation of PV cell connectivity in the cortex[24].

**Short-term administration of Rapamycin rescues long-term loss of PV cell innervation in Tg(*Nkx2.1-Cre*);*Tsc1*flox/+ mice.** So far, our data suggest that *Tsc1* haploinsufficiency in PV cells induces a premature formation of PV perisomatic synapses, which are however not stable and are subsequently lost. We can conceive two mechanisms to explain our observations: (1) Tsc1-mTORC1 signaling plays two distinct, opposing, and age-dependent roles in PV cells, namely, during the first few postnatal weeks, mTORC1 activation promotes PV cell synapse formation, while later it may actively promote synapse pruning, or (2) mTORC1 hyper-activation during an early postnatal phase

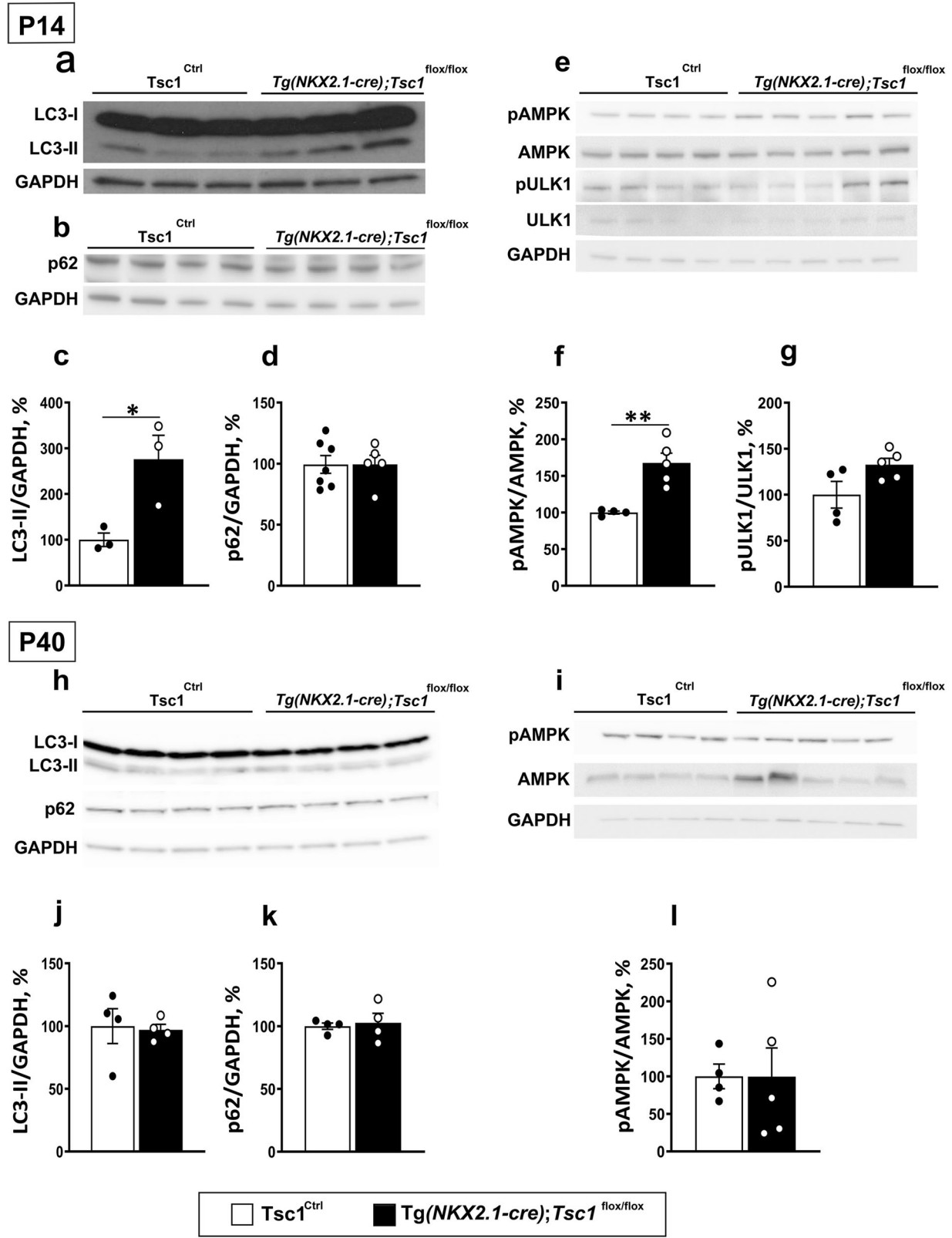

promotes PV cell synapse formation and can cause alterations (such as increased AMPK activation and autophagy) that are directly responsible for the synaptic loss occurring at later ages. To test which one of these two mechanisms is more likely to play a role in the loss of adult PV cell connectivity caused by *Tsc1* deletion, we biolistically transfected PV cells in EP26 cortical organotypic cultures prepared from *Tsc1^flox/flox* mice with GFP

($P_{G67}$-GFP/Cre) or GFP alone (control) and analyzed them at EP34 (Fig. 7). We reasoned that if the first scenario was more likely, then late-onset Tsc1-deletion in PV cells should still cause loss of PV cell innervations. However, our data showed that the innervations formed by $Tsc1^{-/-}$ PV cells are indistinguishable from those formed by age-matched control PV cells for all analyzed parameters, suggesting that TSC1-mTORC1 dysregulation

**Fig. 6 *Tsc1* deletion in GABAergic cells causes transient autophagy dysfunctions in adolescent mice.** Western blot representative bands of LC3-I, LC3-II (**a**), p62 proteins (**b**); pAMPK, AMPK, pULK1, and ULK1 (**e**). **c–g** Quantification reveals that LC3-II and pAMPK/AMPK expressions are higher in P14 Tg (*Nkx2.1-Cre*);*Tsc1^flox/flox^* vs *Tsc1*^Ctrl^ mice (**c**, **f**; Unpaired *t*-test: LC3-II \**p* = 0.0314; pAMPK/AMPK \*\**p* = 0.003), while p62 protein expression was unchanged and pULK1/ULK1 expressions showed a trend towards increased expression (**d**, **g**; Unpaired *t*-test: p62 *p* = 0.9937; pULK1/ULK1 *p* = 0.065). **h**, **i** Western blot representative bands of LC3-I, LC3-II, p62, pAMPK, and AMPK in adult mice and their quantification (**j**, **k**, **l**) show no differences between the two genotypes (Unpaired *t*-test: LC3-II *p* = 0.8446; p62 *p* = 0.7426; pAMPK/AMPK *p* = 0.9925). Molecular weight: LC3-I/II :17/14 kDa; p62: 62 kDa; pAMPK: 62 kDa; AMPK: ~62 kDa; ULK1:140 kDa; pULK1: ~150 kDa; GAPDH: 37 kDa. Number of mice at P14: LC3-II; *Tsc1*^Ctrl^ *n* = 3, Tg(*Nkx2.1-Cre*);*Tsc1^flox/flox^* *n* = 3; p62, pAMPK, AMPK, pULK1, ULK1 *Tsc1*^Ctrl^ *n* = 4, Tg(*Nkx2.1-Cre*);*Tsc1^flox/flox^* *n* = 5. Number of mice at P40: LC3-II and p62; *Tsc1*^Ctrl^ *n* = 4, Tg(*Nkx2.1-Cre*);*Tsc1^flox/flox^* *n* = 4; pAMPK and AMPK; *Tsc1*^Ctrl^ *n* = 4, Tg(*Nkx2.1-Cre*);*Tsc1^flox/flox^* *n* = 5. Data represent mean ± SEM. Source data are provided as a Source Data file.

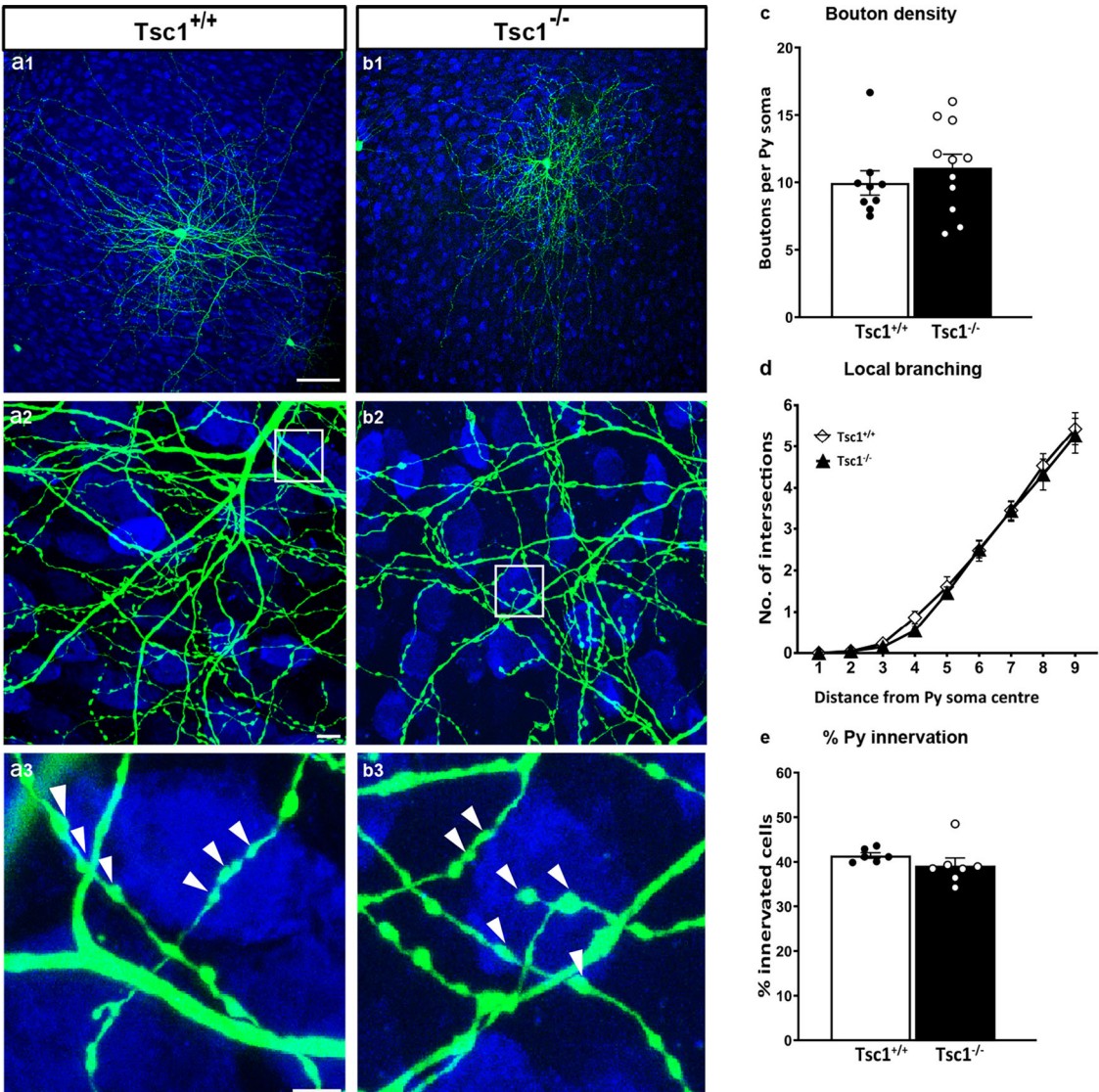

**Fig. 7 Late-onset *Tsc1* deletion in PV cells does not affect their innervation. a1** EP 34 *Tsc1*^+/+^ and **b1** *Tsc1*^−/−^ PV cells show similar axonal branching (**a2**, **b2**) and perisomatic boutons (**a3**, **b3**, arrowheads). **c** Bouton density (Welch's *t*-test, *p* = 0.4091; PV cells: *n* = 9 *Tsc1*^+/+^, *n* = 11 *Tsc1*^−/−^), **d** local branching (Welch's *t*-test, *p* = 0.5789; PV cells: *n* = 9 *Tsc1*^+/+^, *n* = 11 *Tsc1*^−/−^), and **e** percentage of innervation (Welch's *t*-test, *p* = 0.2448; PV cells: *n* = 6 *Tsc1*^+/+^, *n* = 7 *Tsc1*^−/−^) are not significantly different between the two groups. Scale bars: **a1**–**b1**, 50 μm; **a2**–**b2**, 10 μm, and **a3**–**b3**, 5 μm. Data in represent mean ± SEM. Source data are provided as a Source Data file.

before the third postnatal week is likely responsible for the subsequent loss of PV cell connectivity.

This observation raised the possibility that inhibiting mTORC1 hyperactivation during this critical time window might be sufficient to lead to long-term rescue of PV cell connectivity and, possibly, social behavior deficits. To directly test this hypothesis, we first used

an in vitro approach by treating organotypic cultures from heterozygous and homozygous mutant mice with rapamycin (90 ng/ml) from EP10 to EP18 and analyzing PV cell innervation at EP34. In culture from heterozygous mutant mice, rapamycin treatment reversed the decrease in perisomatic bouton density (Supplementary Fig. 9e), terminal axonal branching

(Supplementary Fig. 9f), and percentage of target cell innervations (Supplementary Fig. 9g) caused by *Tsc1* haploinsufficiency. We noted that rapamycin treatment also significantly reduced the percentage of target cells potentially contacted by PV cells in wild-type cultures (Supplementary Fig. 9g).

In cultures from homozygous mutant mice, the same rapamycin treatment only partially reversed the decrease in perisomatic bouton density and terminal axonal branching, while it had not significant effect on the percentage of innervation formed by the mutant PV cells (Supplementary Fig. 10). It is possible that higher rapamycin doses might be required to completely rescue the PV cell innervation phenotype in PV cells from conditional homozygous mice.

Taken together, these data suggest that short-term rapamycin treatment during the early postnatal development can lead to persistent rescue of PV cell connectivity, particularly in case of haploinsufficiency.

Finally, we tested whether short-term rapamycin treatment during early postnatal development can rescue long term effects of *Tsc1* haploinsufficiency in vivo. First, we analyzed PV cell perisomatic synapse innervation in the cortex of P21 *PV-Cre;Tsc1flox/+* mice. Since in our experience, Cre expression does not reach plateau until the fourth postnatal week in the PV-Cre mouse line we used, we generated *PV-Cre;RCE;Tsc1+/+* and *PV-Cre;RCE;Tsc1flox/+* mice, because Cre-dependent GFP expression allowed us to identify perisomatic innervations formed by the PV cells where Tsc1 recombination had likely already occurred (Fig. 8a–f). Consistently to what we observed in mice with embryonic deletion of *Tsc1* in MGE-derived GABAergic cells, postnatal *Tsc1* haploinsufficiency in PV cells caused a premature increase of PV+ perisomatic synapse density (Fig. 8g).

We then treated *PV-Cre*;RCE;*Tsc1flox/+* and their control littermates (*PV-Cre*;RCE;*Tsc+/+*) daily with either rapamycin (3 mg/kg; i.p.) or vehicle from P14 to P21 and analyzed PV cell perisomatic synaptic density at the end of the treatment (Fig. 9a). We found that this treatment was sufficient to rescue the premature PV cell hyper-connectivity (Fig. 9n, o).

Finally, we treated *PV-Cre;Tsc1flox/+*, *PV-Cre;Tsc1flox/flox*, and control littermates daily with either rapamycin (3 mg/kg; i.p.) or vehicle from P14 to P21 and analyzed PV cell perisomatic synaptic density and social behavior at P45 (Fig. 10a). We found that rapamycin treatment restricted during this specific early developmental time window was sufficient to rescue the density of PV+/Geph+ puncta to wild-type levels in both hetero and homozygous mutant, adult mice (Fig. 10b–i). In contrast, rapamycin treatment completely rescued social behavior deficits (both social approach and social novelty preference) in conditional heterozygous (*PV-Cre; Tsc1flox/+*) but not in conditional homozygous mice (*PV-Cre; Tsc1flox/flox*) (Fig. 10l, m).

Since PV is also expressed in Purkinje and granule cells in the cerebellum and Purkinje cell-specific deletion of *Tsc1* has been shown to cause social behavioral deficits[38], we looked at cerebellum cyto-architecture, by immunolabeling cerebellar slices from vehicle- and rapamycin-treated mice with Calbindin, PV, and NeuN (Supplementary Fig. 11). While we did not observe any obvious abnormalities in *PV-Cre;Tsc1flox/+* mice, the cerebellar cellular anatomy of *PV-Cre;Tsc1flox/flox* mice was significantly altered. In particular Purkinje cell numbers were greatly reduced and their dendritic arbors severely abnormal, resembling those observed during the first postnatal week. These abnormalities were only partially rescued by rapamycin treatment

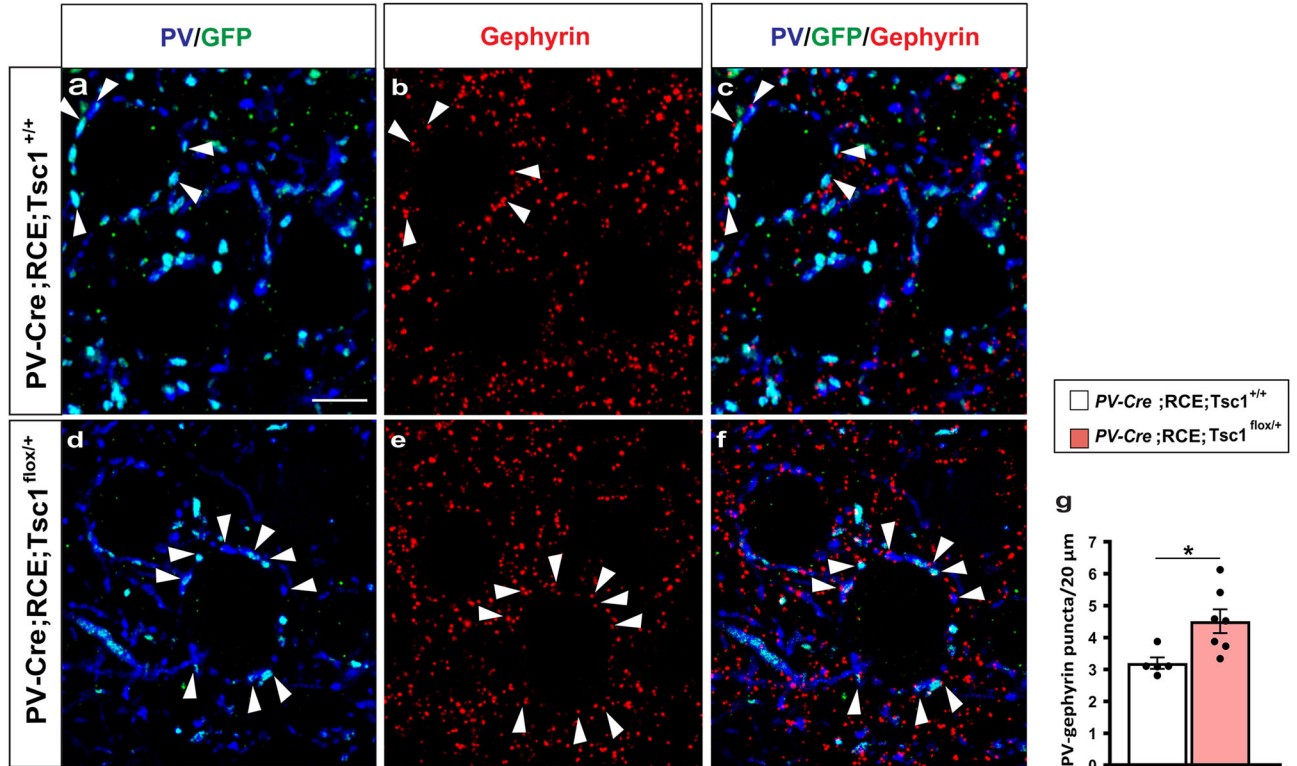

**Fig. 8 Postnatal onset of *Tsc1* haploinsufficiency in PV cells induces premature putative synapse formation. a–f** Representative immunostained sections of somatosensory cortex labeled for PV/GFP (blue/green) and gephyrin (red) in P22 *PV-Cre;RCE;Tsc1+/+* (**a–c**) and *PV-Cre;RCE;Tsc1flox/+* mice (**d–f**). White arrowheads indicate PV/Gephyrin colocalized puncta (**c** and **f**). Scale bar: 5 µm. **g** Quantification of PV/GFP-Gephyrin colocalized puncta (Welch's *t*-test, *$*p = 0.0121$). Number of mice: $n = 5$ for *PV-Cre;RCE;Tsc1+/+* and $n = 7$ for *PV-Cre;RCE;Tsc1flox/+*. Data represent mean ± SEM. Source data are provided as a Source Data file.

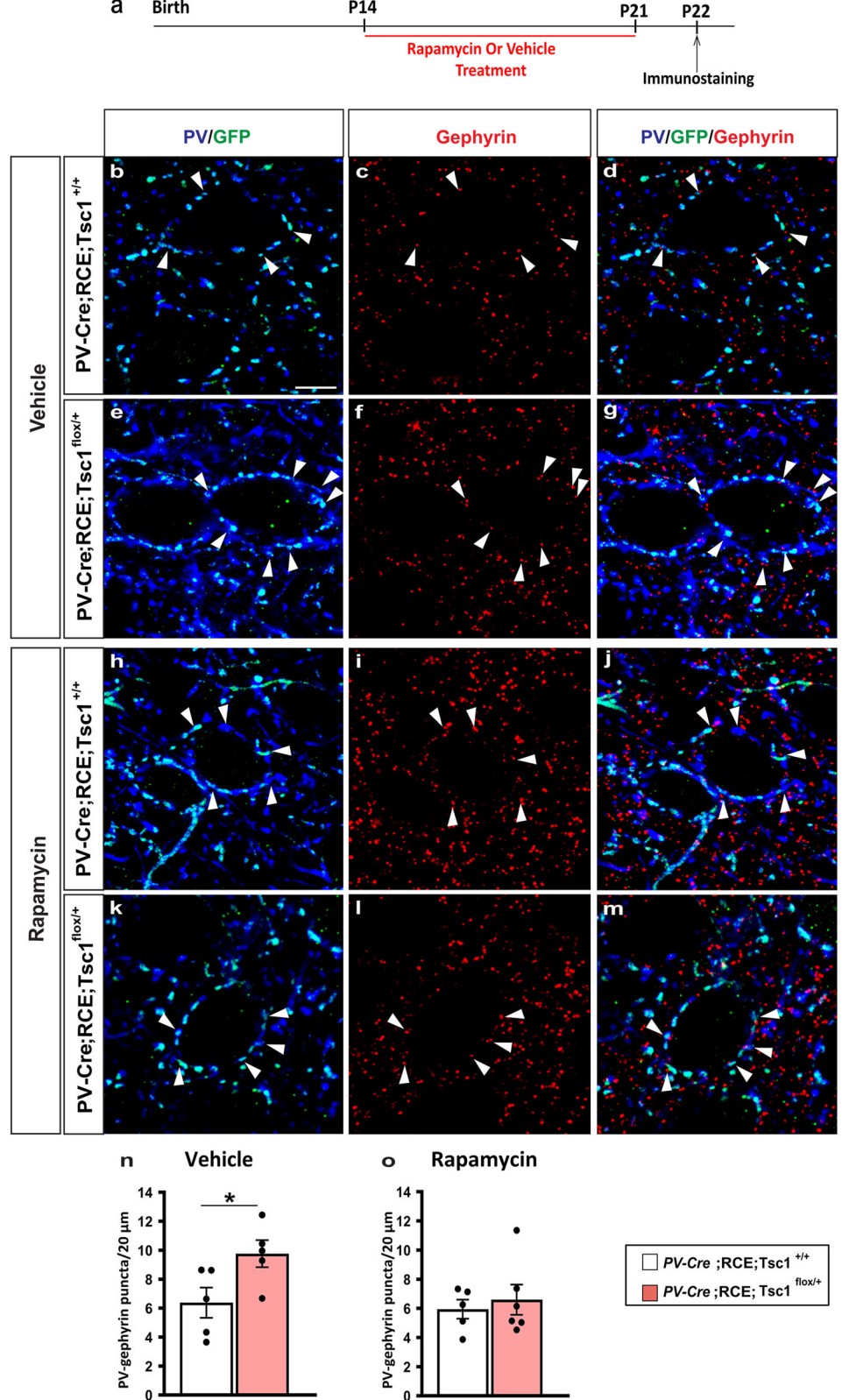

**Fig. 9 Short term Rapamycin treatment rescues the premature increase of PV cell perisomatic innervation caused by *Tsc1* haploinsufficiency. a** Schematic of the treatment paradigm. **b–m** Representative immunostained sections of somatosensory cortex labeled for PV/GFP (blue/green) and gephyrin (red) in Vehicle (**b–g**) and Rapamycin (**i–m**) treated P22 mice showing PV/GFP-Gephyrin colocalized boutons (arrowheads) (**d, g** and **j, m**). Scale bar: 5 μm. **n, o** PV/GFP-Gephyrin colocalized puncta in Vehicle-treated (**n**, Welch's *t*-test, *$p = 0.0423$) and Rapamycin-treated mice (**o**, Welch's *t*-test, $p = 0.6088$). Vehicle treatment: $n = 5$ for both *PV-Cre;RCE;Tsc1$^{+/+}$* and *PV-Cre;RCE;Tsc1$^{flox/+}$* mice. Rapamycin treatment: $n = 5$ *PV-Cre;RCE;Tsc1$^{+/+}$*, $n = 6$ *PV-Cre;RCE;Tsc1$^{flox/+}$* mice. Data represent mean ± SEM. Source data are provided as a Source Data file.

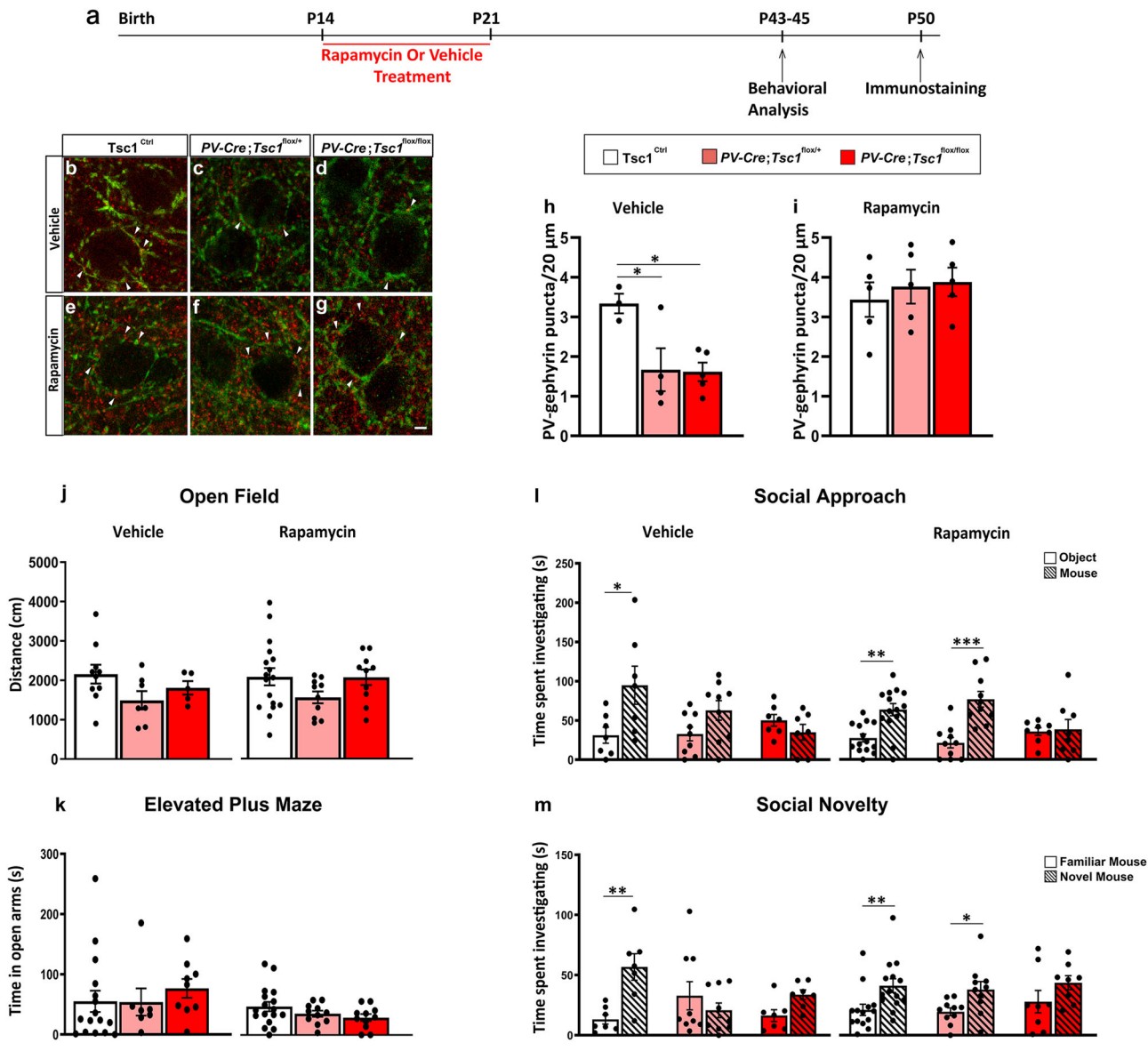

**Fig. 10 Short term rapamycin treatment rescues loss of PV cell connectivity and social behavior deficits in adult heterozygous mutant mice. a** Schematic for treatment paradigm. **b–g** Representative immunostained sections of somatosensory cortex labeled for PV (green) and gephyrin (red) in Vehicle (**b–d**) and Rapamycin (**e–g**) treated mice showing PV-Gephyrin colocalized boutons (arrowheads). **h** PV-Gephyrin colocalized puncta in Vehicle (one-way Anova, *p = 0.036; Tukey's multiple comparisons test: Tsc1^Ctrl vs PV-Cre; Tsc1^flox/+ *p = 0.0401; Tsc1^Ctrl vs PV-Cre; Tsc1^flox/flox *p = 0.0281, PV-Cre; Tsc1^flox/+ vs PV-Cre;Tsc1^flox/flox *p = 0.9942) and **i** Rapamycin treated mice (one-way Anova, p = 0.7355; Tukey's multiple comparisons test: Tsc1^Ctrl vs PV-Cre;Tsc1^flox/+ p = 0.8421; Tsc1^Ctrl vs PV-Cre; Tsc1^flox/flox p = 0.7298, PV-Cre; Tsc1^flox/+ vs PV-Cre; Tsc1^flox/flox p = 0.9774). Number of vehicle-treated mice: Tsc1^Ctrl, n = 3; PV-Cre; Tsc1^flox/+, n = 4; PV-Cre; Tsc1^flox/flox, n = 5. Number of rapamycin treated mice: n = 5 for all the genotypes. **j** Open field test. Vehicle-treated mice: Tsc1^Ctrl, n = 10; PV-Cre;Tsc1^flox/+, n = 7; PV-Cre;Tsc1^flox/flox, n = 5. Rapamycin-treated mice: Tsc1^Ctrl, n = 17; PV-Cre;Tsc1^flox/+, n = 10; PV-Cre;Tsc1^flox/flox, n = 10. **k** Elevated plus maze test. Vehicle-treated mice: Tsc1^Ctrl, n = 16; PV-Cre; Tsc1^flox/+, n = 7; PV-Cre;Tsc1^flox/flox, n = 9; Rapamycin-treated mice: Tsc1^Ctrl, n = 17; PV-Cre;Tsc1^flox/+, n = 11; PV-Cre;Tsc1^flox/flox, n = 11. **l, m** Rapamycin treatment from P14-21 rescues social approach (**l**, two-way ANOVA vehicle, $F_{genotype}$ (2, 20) = 1.615 p = 0.2238, $F_{time}$ (1, 20) = 4.789 p = 0.0407, $F_{genotype*time}$ (2, 20) = 3.395 p = 0.0538; *p = 0.0230, Holm–Sidak's multiple comparisons test. Two-way ANOVA rapamycin, $F_{genotype}$ (2, 29) = 1.027 p = 0.3709, $F_{time}$ (1, 29) = 22.08 p < 0.0001, $F_{genotype*time}$ (2, 29) = 4.580 p = 0.0187; **p = 0.0019, ***p = 0.0002, Holm–Sidak's multiple comparisons test) and social novelty deficits (**m**, two-way ANOVA vehicle, $F_{genotype}$ (2, 20) = 0.8281 p = 0.4513, $F_{time}$ (1, 20) = 5.876 p = 0.0250, $F_{genotype*time}$ (2, 20) = 6.036 p = 0.0089; **p = 0.0053, Holm–Sidak's multiple comparisons test. Two-way ANOVA rapamycin, $F_{genotype}$ (2, 29) = 0.4667 p = 0.6317, $F_{time}$ (1, 29) = 18.84 p = 0.0002, $F_{genotype*time}$ (2, 29) = 0.09351 p = 0.9110; **p = 0.0081, *p = 0.0336, Holm–Sidak's multiple comparisons test) in PV-Cre; Tsc1^flox/+, but not in PV-Cre;Tsc1^flox/flox mice. Vehicle-treated mice: Tsc1^Ctrl, n = 7; PV-Cre; Tsc1^flox/+, n = 9; PV-Cre;Tsc1^flox/flox, n = 7. Rapamycin-treated mice: Tsc1^Ctrl, n = 14; PV-Cre;Tsc1^flox/+, n = 10; PV-Cre; Tsc1^flox/flox, n = 8. Scale bar: 5 μm. Data represent mean ± SEM. Source data are provided as a Source Data file.

from P14-21. Thus, cerebellar impairments might contribute to the lack of rescue of social behavior deficits by early rapamycin treatment in the conditional homozygous mice[38,45].

In summary, taken together, these data suggest that short-term rapamycin treatment during a critical postnatal window has long-lasting protective effects on GABAergic connectivity and social

behavior in the context of *Tsc1* haploinsufficiency, which is typical of TSC patients.

## Discussion

GABAergic circuits play a central role in the social behaviors affected in mTORC1-related neurodevelopmental disorders[29,34,46]. Here, we used single cell genetic manipulation approaches and genetic mouse models to investigate how dysregulation of mTORC1 signaling affects the development and maintenance of cortical PV GABAergic cells. We found that PV cell-specific haplosinsufficiency of *Tsc1*, a key negative regulator of mTORC1 signaling, leads to reduced PV cell connectivity in adult mice, with mutant PV cells forming less and smaller synapses. Mice either haploinsufficient or lacking *Tsc1* in PV cell exhibit altered social behavior. The fact that the conditional heterozygous mice exhibited comparable social behavior dysfunction as the conditional homozygous mice suggest that these findings are relevant for TSC, which is an autosomic dominant disorder.

Based on our data, we hypothesize that the loss of PV cell connectivity in adult mice is dependent on the premature formation of PV cell innervations during a critical, developmental period. First, single-PV cell *Tsc1* deletion with onset at EP10 caused a premature formation of PV cell innervation by EP18, followed by excessive synaptic pruning, while *Tsc1* deletion with onset at EP26, after PV cell innervations are stabilized, did not caused any changes in perisomatic innervations. In particular, the effects of *Tsc1* deletion in single, sparse PV cells in otherwise wild-type organotypic cultures suggest that *Tsc1* acts in a cell autonomous fashion to regulate PV cell innervation. Second, 1-week rapamycin treatment from EP10–18 was sufficient to protect long-term PV cell innervation from excessive pruning. Third, consistent with the data in vitro, deletion or haploinsufficiency of *Tsc1* in vivo, with either embryonic (Tg(*Nkx2.1-Cre*);*Tsc1^{lox}* mice) or postnatal (*PV-Cre;RCE;Tsc1^{lox}* mice) onset, leads to premature increases of PV cell perisomatic innervations in preadolescent mice followed by a significant loss in adults. On the other hand, in vivo rapamycin treatment during a restricted, sensitive period (P14–22) rescued both the premature hyper-connectivity phenotype at P22 and the loss of PV cell perisomatic synapses in conditional heterozygous adult mice compared to wild-type littermates. Fourth, conditional mutant *Tsc1* mice showed altered autophagy-associated processes at P14, when PV cells are at the peak of their maturation phase[24,26,27,47] but not at 6 postnatal weeks, when PV cell connectivity has already reached maturity.

Multiple, parallel cellular mechanisms most likely underlie the altered developmental time course of PV cell connectivity. *Tsc1* deletion-mediated mTORC1 hyperactivity may promote growth, via increased protein synthesis[2,48]. In addition, mTORC1 activation has been shown to affect autophagy. Importantly, autophagy is mechanistically distinct in neurons compared to dividing cells. In fact, most studies on mTORC1 function, which used dividing cells, concluded that mTORC1 activation inhibits autophagy. On the other hand, Di Nardo and colleagues showed that *Tsc1/2*-deficient neurons displayed increased autophagic activity, which was dependent of AMPK activation[42]. Our results are consistent with these findings, since we observed increased AMPK phosphorylation and LC3-II levels, and a trend towards increased ULK1 phosphorylation, in *Tsc1*-deficient GABAergic cells in the olfactory bulb in vivo. Compromised autophagy, and accumulation of defective organelles, for example mitochondria, may be one of the downstream causes of PV cell axonal loss and synaptic pruning[49–51]. In addition, AMPK plays a critical role in fine tuning short-term plasticity and maintaining prolonged

synaptic efficacy by regulating mitochondria recruitment and positioning to presynaptic sites[52] and it is possible that altered levels of AMPK activation may contribute to synaptic dysfunction during intensive synaptic activity. Further studies are needed to address the specific effects of AMPK hyper-phosphorylation in *Tsc1* haploinsufficient, cortical PV cells.

A recent study showed that *Tsc1* deletion specifically in somatostatin (SST)-expressing GABAergic interneurons leads to altered firing properties of a percentage of cortical SST neurons in both conditional heterozygous and homozygous mice but to overall reduced synaptic output only in the conditional homozygous mutants[12], suggesting that the formation and refinement of PV cell synaptic connectivity is more sensitive to *Tsc1* haploinsufficiency than SST neurons even if they both originate from the MGE. Another group generated *PV-Cre;Tsc1* conditional knockout mice, but in contrast to our findings, found no physiological phenotypes[11], thus concluding that most TSC phenotypes arise from excitatory pyramidal neurons. One possible explanation for this discrepancy is that in this study electrophysiological analysis was performed at P28–P30, which is after the phase of premature synapse formation but before synaptic loss might become detectable in Tsc1 haploinsufficient PV cells. Consistent with this hypothesis, we did not observe significant differences in perisomatic innervations formed by mutant Tsc1 PV cells in organotypic cultures at EP24 (Supplementary Fig. 5).

Our data suggest the existence of a sensitive period, namely a time window during which therapy is effective for the treatment of a specific phenotype[53], for the treatment of social behavior impairments caused by *Tsc1* haploinsufficiency in PV cells. Strikingly, treatment limited to 1 week in preadolescent mice (P14–22) was sufficient to rescue both cortical PV cell innervation and social behavior deficits in adult *PV-Cre;Tsc1^{lox/+}* mice. On the other hand, rapamycin treatment was sufficient to rescue cortical PV cell connectivity but not behavioral deficits in homozygous mutant mice, most likely due to the persistence of cerebellar defects. In fact, work from the Sahin's group showed that *Tsc1*-deletion or haploinsufficiency in Purkinje cells was sufficient to cause autistic-like phenotypes in mice[38]. Interestingly, these cerebellar-dependent social behavior phenotypes could be rescued by continuous rapamycin treatment initiated either at P7[38] or at 6 postnatal weeks[45] in homozygous, Purkinje cell-specific mutant mice. It is possible that the sensitive period to ameliorate the deficits of cell survival and excitability caused by *Tsc1* deletion in Purkinje cells may be well into adolescence, later than that sufficient to rescue the deficits in cortical GABAergic PV interneurons. Alternatively, the chronic presence of rapamycin might be needed to inhibit the physiological changes in adult mutant Purkinje cells.

Multiple brain regions and neuronal circuits likely contribute to the different cognitive tasks, which underlie social behaviors. Cortical PV cell activity modulates sensory responses[17,46], which are required for the development of normal social interaction behaviors[54]. Postmortem analysis of brains from ASD patients as well as animal models for ASD (such as Mecp2 and Shank3 mutants) revealed abnormalities in PV cell circuits in multiple brain regions, including primary sensory cortices[29–33,55–61]. Targeting PV cell circuit impairments might thus be a rational approach to ameliorate social interaction problems, however the developmental and cellular processes that lead to PV cell dysfunction are likely dependent of the underlying etiology.

Our results suggest that *Tsc1* haploinsufficiency in PV cells leads to defects in adult connectivity that can be rescued by targeted treatment during a well-defined postnatal sensitive period. Multiple proteins in mTOR signaling pathway are either high confidence ASD-causative genes or underlie disorders with high ASD comorbidity[62], therefore highlighting this pathway as a

possible etiological hub for the disorder. Whether a similar altered developmental trajectory of PV cell circuit maturation might be common to different mTORpathies remains to be explored. Interestingly, a recent study by Thion and collaborators reported that two different prenatal immune challenges lead to premature maturation of PV cell connectivity followed by reduced PV cell inhibitory drive in adult somatosensory cortex, similarly to what we observed[63]. Prenatal inflammation has been associated with etiology of several neuropsychiatric disorders, including ASD[64]. It will be interesting to investigate whether maternal immune activation affects PV circuit development by impinging on the Tsc/mTOR pathways.

Finally, while rapamycin, a mTORC1 blocker, was effective in rescuing PV cell connectivity and social behavior deficits in *PV-Cre;Tsc1lox/+* mice, we cannot exclude that *Tsc1* deletion might have additional effects independent of mTORC1 signaling, since mice carrying hyperactive mTORC1 in Purkinje cells were recently reported to display different behavioral alterations compared to Purkinje-cell specific Tsc1-lacking mice[65]. A better understanding of the complexity of mTORC1 signaling network regulation, feedback and compensatory loops, may lead to the discovery of new molecular drug targets.

The mTOR inhibitor everolimus is approved by FDA for the treatment of subependymal giant cell astrocytomas, angiomyoli-pomas, and complex partial seizures in TSC patients. Several recent studies have addressed whether everolimus treatment could have positive effects on cognition and autistic behavior in TSC patients, but have so far produced controversial results[66–71]. One important point is that children younger than 4 years old were excluded from these studies. Our data suggests that an early age of onset of the treatment might be critical to improve specific cognitive and behavioral long term outcomes.

## Methods

**Animals.** Tsc1 floxed mice with loxP sites flanking exons 17 and 18 of Tsc1 gene (*Tsc1flox/flox*) were purchased from Jackson Laboratories (Cat# 005680). Two separate driver mouse lines expressing Cre recombinase, (1) Tg(*Nkx2.1-Cre*)[41], (Jackson Laboratories, Cat# 008661) and (2) *PV-Cre* (Jackson Laboratories, Cat# 008069)[72] were crossed to the Tsc1 floxed mice and the respective progenies were backcrossed to generate the heterozygous, homozygous, and control genotypes within the same litter. To control the pattern of expression of Cre, we introduced the RCE allele using Gt(ROSA)26Sortm1.1(CAG-EGFP)Fsh/J mice (Jackson laboratories). The RCE line carries a loxP-flanked STOP cassette upstream of eGFP sequence within the Rosa26 locus. Removal of the loxP-flanked STOP cassette by Cre-mediated recombination allows promoter-specific downstream eGFP expression[73]. All mice were housed under standard pathogen-free conditions in a 12 h light/dark cycle, at 21 °C and 40% humidity, and with ad libitum access to sterilized laboratory chow diet. Animals were treated in accordance with Canadian Council on Animal Care and protocols were approved by the Comité institutionnel des bonnes pratiques animales en recherche (CIBPAR) of CHU Ste-Justine Research Center.

**Mice genotyping.** DNA was extracted from mouse tails and genotyped to detect the presence of Cre alleles and Tsc1 conditional and wild-type alleles. Polymerase chain reaction (PCR) was performed using the primers listed in Supplementary Table 1. Three separate primers were used for detecting TSC1 alleles with band sizes of 295 bp for the wild-type and 480 bp for the floxed allele. Three separate primers were also used for detecting Cre in the Tg(Nkx2.1-Cre) breeding; which generated 550 and 220 bp (mutant and wild-type) bands. Primers for detecting Cre in PV-Cre generated 400 and 526 bp (mutant and wild-type) bands. To detect the presence of RCE alleles, three separate primers were used which generated 350 and 550 bp bands.

**Slice culture and biolistic transfection.** Slice culture preparation was done as described previously[24]. Postnatal day 4 or 5 (P4 or P5) mouse pups were decapitated, and brains were rapidly removed and immersed in ice-cold culture medium (DMEM, 20% horse serum, 1 mM glutamine, 13 mM glucose, 1 mM $CaCl_2$, 2 mM $MgSO_4$, 0.5 μm/ml insulin, 30 mM HEPES, 5 mM $NaHCO_3$, and 0.001% ascorbic acid). Coronal brain slices obtained starting from the occipital cortex until the end of the somatosensory cortex, 400 μm thick, were cut with a Chopper (Stoelting, Wood Dale, IL). Slices were then placed on transparent Millicell membrane inserts (Millipore, Bedford, MA), usually three to four slices/insert, in 30 mm Petri dishes

containing 0.75 ml of culture medium. Finally, the slices were incubated in a humidified incubator at 34 °C with a 5% $CO_2$-enriched atmosphere and the medium was changed three times per week. All procedures were performed under sterile conditions. Constructs to be transfected were incorporated into "bullets" that were made using 1.6 μm gold particles coated with a total of ~50 μg of the DNA(s) of interest. These bullets were used to biolistically transfect slices by Gene gun (Bio-Rad, Hercules, CA) at high pressure (180 Ψ). In order to delete *Tsc1* in single PV cells in an otherwise wild-type background, we transfected organotypic slices from *Tsc1flox/flox* mice either with $P_{G67}$-GFP (*Tsc1+/+*, control PV cells) or $P_{G67}$-Cre/$P_{G67}$-GFP (*Tsc1−/−* PV cells). Organotypic cultures from Tg(*Nkx2.1-Cre+/−*; *Tsc1flox/+*), Tg(*Nkx2.1-Cre+/−*;*Tsc1flox/flox*) and *Tsc1Ctrl* were transfected with $P_{G67}$-GFP to visualize PV cells. For each experimental group, cortical slices were prepared from at least three mice. The majority of neurons labeled by using the $P_{G67}$ promoter were PV-positive cells[24,25,74], while a minority (~10%) were pyramidal cells. Pyramidal cells were recognized by the complexity of their dendritic arbor, including an apical dendrite, and the presence of numerous dendritic spines. PV immunolabeling (see protocol below) was performed to confirm PV cell identity before imaging.

**Immunohistochemistry.** Mice were perfused transcardially with saline followed by 4% paraformaldehyde (PFA 4%) in phosphate buffer (PB 0.1 M, pH 7.2). Brains were post-fixed with 4% PFA overnight and subsequently transferred to a 30% sucrose solution in sodium phosphate-buffer (PBS) for 48 h. They were then frozen in molds filled with Tissue Tek using a 2-Methylpentane bath cooled with a mixture of dry ice and ethanol (~ −70 °C). Optimal cutting temperature and coronal sections of 40 μm or horizontal sections (40–50 μm for the olfactory bulb) were obtained using a cryostat (Leica VT100). Organotypic cultures were fixed overnight at 4 °C in 4% PFA in PB 0.1 M, pH 7.2, then washed in PBS, incubated in 30% sucrose/PBS, and subjected to a freeze/thaw cycle at −20 °C. Brain sections or organotypic cultures were blocked in 10% normal goat serum (NGS) and 1% Triton X-100 for 2 h at room temperature (RT). Slices were then incubated for 48 h at 4 °C with the following primary antibodies: rabbit anti-phospho-S6 (1:1000, Cell Signaling, Cat# 5364), mouse anti-NeuN (1:400, Millipore, Cat# MAB377), chicken anti-NeuN (1:500, Millipore, Cat# ABN91), mouse anti-PV (1:1000, Swant, Cat# 235), rabbit anti-PV (1:1000, Swant, Cat# PV27), guinea pig anti-PV (1:1000, Synaptic Systems, Cat# 195004), mouse anti-gephyrin (1:500, Synaptic Systems, Cat# 147021), mouse anti-Calbindin (1:1000, Abcam, Cat# 9481), rabbit anti-VGAT (1:1000, Synaptic System, Cat #131003), and chicken anti-GFP (1:1000, Abcam, Cat# 13970). It was followed by incubation with secondary antibodies for 2 h at RT to visualize primary antibodies. The secondary antibodies used were Alexa-Fluor conjugated 488, 555, 594, 633, and 647 (1:400, Life technologies; 1:1000, Cell Signaling Technology). Olfactory bulb sections were stained with DAPI (Life Technologies, Cat# D3571). After rinsing in PBS (three times), the slices were mounted in Vectashield mounting medium (Vector).

**Confocal imaging and quantitative analysis.** All imaging was performed using Leica confocal microscopes (SPE, SP8, or SP8-STED). For PV cell innervation analysis, PV cells were first imaged using 10x (NA0.4) to record the overall cell morphology, and then multiple (2–3) stacks of their axonal fields were acquired using a glycerol immersion 63x (NA1.3) objective at 0.5 or 1 μm z-step in the first 150 μm from the PV cell soma. The complexity of the PV axon branches around a pyramidal cell soma was reported as the average number of intersections, bouton density, and percentage of innervated pyramidal cells. The number of intersections represented the intersections between a basket cell axon and the Sholl spheres (9 μm, increment of 1 μm) from the center of the pyramidal cell soma. Bouton density around each basket cell represented the total number of GFP+ boutons in a radius of 9 μm from the center of the pyramidal cell soma. About 12–24 pyramidal cells were analyzed for each basket neuron. To determine the percentage of pyramidal innervated by basket cells axon, we quantified the number of pyramidal cells soma that were contacted by the GFP+ axon and divided the later by the total number of somata in a confocal stack field.

For in vivo analysis, we imaged somatosensory cortex layers 2/3 and 5/6 using 20X (NA 0.75) and 63X oil (NA1.3) objectives. The 20X objective was used to acquire images for analyzing the percentage of PV/pS6 cellular colocalization, PV+/NeuN+ cell density, GFP+/GFP+ PV+ cell density (specificity of PV recombination) and PV+/GFP+ PV+ cell density (recombination rate). For the analysis of perisomatic innervation 63x glycerol objective was used to acquire images for quantifying PV and gephyrin puncta. At least three confocal stacks from three different brain sections were acquired in layers 2/3 and 5/6 of somatosensory cortex with z-step sizes of 0.5 (for synapse quantification) or 1 μm (for cell density quantification). Cell soma size and cell density were quantified using Neurolucida (MBF Software). Fluorescence intensity of pS6 signal in PV cells was calculated using ImageJ and LAS X (Leica Application Suite X) software. In LAS X, 8 to 10 cells were chosen on various focal plane and encircled by using the polygon tool. This process generated the mean gray values of each cell. On the other hand, the mean gray values of four spots without any staining from the same focal plane were used as background, which were removed in order to normalize the data. For each animal, three sections were used in order to minimize the variability across the different groups. PV+, gephyrin+, and PV+/gephyrin+ puncta were counted

around NeuN positive somata after selecting the confocal plane with the highest soma circumference using Neurolucida and ImageJ-Fiji softwares. At least 6–10 NeuN positive somata were selected in each confocal stack.

To analyze olfactory bulb, we imaged the EPL of adult (~2 months) mice of both sexes using a 100X (NA 1.4, oil immersion) objective, zoom 1.5. We analyzed four sections per animal, by choosing four to five regions of interest (ROI, $30 \times 30$ $\mu m^2$) in each of the confocal stack and quantifying VGAT+, gephyrin+ puncta using ImageJ-Fiji software. Investigators were blind to the genotypes during the analysis.

**Electron microscopy**. The electron microscopy was carried out on two groups, $Tsc1^{Ctl}$ and $PV\text{-}Cre;Tsc1^{flox/+}$ mice, at P60. Mice were anesthetized and perfused with 0.1 M PBS (0.9% Nacl in PB 0.2 M; pH 7.4) followed by 2.5% glutar-aldehyde + 2% PFA in 0.1 M PB, pH 7.4. Following perfusion, the brains were further fixed for 2 h at RT in the perfusion solution. Transverse 50-μm-thick sections of the brain were cut in cooled PBS with a vibratome (Leica, VT1000S). They were stored at −20 °C in antifreeze solution (40% PB, 30% ethylene glycol, and 30% glycerol) until used. Sections were immersed in 0.1% borohydride (in PBS) for 15 min at RT, washed in PBS, and processed freely floating following a pre-embedding immunoperoxidase protocol previously described[75]. Briefly, after rinsing in PBS, sections were preincubated (1 h) at RT in a protein blocking solution (Expose Rabbit-Specific HRP/DAB detection IHC Kit, Abcam, Cambridge, UK, ab80437). Then, the sections were incubated for 48 h at 4 °C with rabbit anti-PV (1:1000, Swant, Cat# PV27) in PBS containing 1% NGS, followed by wash (three times in PBS) and incubation for 45 min at RT, in goat anti-rabbit horseradish peroxidase (HRP) conjugate (Abcam, Expose Kit, Cat# ab80437). After rinsing in PBS, immunoreactivity was visualized with hydrogen peroxide in the presence of di-aminobenzodine (DAB Chromogen, Abcam Expose Kit, Cat# ab80437). Thereafter, sections were rinsed in PB, postfixed flat in 1% osmium tetroxide for 1 h and dehydrated in ascending concentrations of ethanol (50%, 70%, 90%, 100%, and finally in ethanol anhydrous). They were then treated with propylene oxide and then impregnated in resin overnight (Durcupan ACM; Sigma) at RT, mounted on aclar embedding film (EMS, Hatfield, PA) and cured at 55 °C for 48 h. Areas of interest from the somatosensory cortex (layers 5/6) were excised from the embedded sections and glued to the tip of prepolymerized resin blocks. Ultrathin (50–70 nm) sections were cut with an ultramicrotome (Reichart UltracutS, Leica, Wetzlar, Germany), collected on bare 150 square-mesh copper grids (Electron Microscopy Sciences, Hatfield, PA), stained with lead citrate, and examined at 80 KV with a Philips CM100 electron microscope, equipped with an 8 MB digital camera (AMT XR80).

To analyze the electron microscopy data, cellular profiles were identified according to well established criteria[75]. All PV labeled structures were classified in different categories such as: dendritic shafts, axons, and axon terminals. All the subcellular profiles that were difficult to identify were classified as "unknown". To provide a better appraisal of the frequency of each type of cellular elements displaying immunolabelling, about 80 to 100 micrographs were randomly taken at 25,000X in each animal, corresponding to a total surface of ~2000 $\mu m^2$. Labeled profiles were counted in all micrograph. Results were expressed as number of immunopositive profiles per 100 $\mu m^2$ of neuropil then normalized over results from the control mice. The area of neuropil and synapses lengths were measured using Neurolucida (MicroBrighField).

**Western blot**. Whole lysate proteins were extracted from the olfactory bulb where GABAergic cells are highly enriched. The olfactory bulbs of $Tsc1^{Ctrl}$ and Tg ($Nkx2.1\text{-}Cre$);$Tsc1^{flox/flox}$ mice were dissected at P14 and P40 and snap frozen in liquid nitrogen. The tissue was then incubated in lysis buffer (150 mM sodium chloride, 1% Triton x-100, 0.5% sodium deoxycholate, 0.1% SDS, 50 mM TrisHCl, pH 8, 2 mM EDTA supplemented with a protease inhibitor cocktail III (Calbiochem)). The concentration of total protein was measured using the Bradford assay (BioRad). Proteins were separated on Novex Tris-Glycine 16% or NuPage Bis-Tris 4–12% protein gels (Invitrogen) in SDS running buffer and were transferred to PVDF membranes (BioRad). The following primary antibodies were used: rabbit anti-LC3B (1:1000, Novus, Cat #NB100-2220), rabbit anti-p62 (1:500, Proteintech, Cat# 18420-1-AP), rabbit anti-pAMPK (1:800, T172, Cell Signaling, Cat# 2535), rabbit anti-AMPK (1:1000, Cell Signaling, Cat#2532), rabbit anti-ULK1 (1:1000, D8H5; Cell Signaling, Cat# 8054), rabbit anti-pULK1 (1:1000, Ser555, D1H4; Cell Signaling, Cat# 5869), and mouse anti-GAPDH (1:5000, ThermoFisher, Cat# MA5-15738). Bands were quantified using Image J software. The intensity of LC3 and p62 bands was normalized over the intensity of the GAPDH band. pAMPK and pULK1 bands were further normalized over total AMPK and ULK1 levels, respectively.

**Rapamycin treatment**. For in vitro experiments, organotypic cultures were prepared from Tg($Nkx2.1\text{-}Cre$);$Tsc1^{flox}$ mice and were treated with Rapamycin from equivalent postnatal day 10 (EP10) to EP18. Rapamycin (90 ng/ml, LC Laboratories, Woburn, MA, USA) was dissolved in the culture medium, which was changed every 48 h. For each mouse, half of the organotypic cultures were treated with rapamycin while the other half remained in regular culture medium, hence allowing us to have internal controls.

For in vivo treatment, rapamycin was administered daily (3 mg/kg; i.p.) to $PV\text{-}Cre;Tsc1^{flox}$ pups from P14 to P21. Rapamycin stock solution (20 mg/ml in 100% ethanol) was stored at −20 °C. Before injection, stock solution was diluted in 5% Tween 80 and 5% polyethylene glycol 400 to a final concentration of 1 mg/ml rapamycin in 4% ethanol[76].

**Mouse behavior tests**. Investigators were blind to genotype during both testing and analysis. Mice of both sexes were used in all experiments.

*Open field*. A mouse was placed at the center of the open-field arena and the movement of the mouse was recorded by a video camera for 10 min. The recorded video file was later analyzed with the SMART video tracking system (v3.0, Harvard Apparatus). To measure exploratory behavior, total distance traveled during the 10 min period, and the time spent in the center versus the periphery was calculated. The open field arena was cleaned with 70% ethanol and wiped with paper towels between each trial.

*Elevated plus maze*. The apparatus consists of two open arms without walls across from each other and perpendicular to two closed arms with walls joining at a central platform. A mouse was placed at the junction of the two open and closed arms. Time spent in the open versus closed arms was video recorded for 5 min. Recordings were scored to measure time spent in open arms, closed arms, and center regions, respectively.

*Three chamber social approach and social novelty tests*. Mice (P45–60) were placed in the middle of the central chamber and allowed to explore all the chambers for 10 min for habituation. After habituation, a wire cage containing an unfamiliar conspecific of the same sex and age (Stranger 1) was placed inside one chamber, while an empty wire cage was placed in the second chamber. Mice were allowed to freely explore the three chambers of the apparatus for 10 min. Social approach was evaluated by quantifying the time spent by the test mice with the object or the mouse in each chamber during the 10 min session. At the end of 10 min, a new unfamiliar mouse of the same sex and age (Stranger 2) was placed in the previously unoccupied wire cage and the test mouse observed for an additional 10 min to assess social novelty. Social novelty was evaluated by quantifying the time spent by the test mouse with either the familiar mouse (Stranger 1) or the newer mouse (Stranger 2) in each chamber during the third 10 min session. Strangers 1 and 2 originated from different home cages and had never been in physical contact with the test mice or with each other. Mice that stayed for the full 10 min session in only one chamber were excluded from the analysis.

**Statistics and reproducibility**. All experiments were repeated independently at least two times. All the statistical analyses were performed using Prism 7.0 (GraphPad Software). Prior to making comparisons across values, the normality of distribution was tested using D'Agostino–Pearson test. Differences between two experimental groups was assessed using two-tailed $t$-test or $t$-test with Welch's correction (for small sample size) for normally distributed data and Mann–Whitney test for not normally distributed data. Differences between three or more experimental groups were assessed with one-way ANOVA and post hoc comparison. For non-normally distributed data, nonparametric Kruskal–Wallis one-way ANOVA test was used. In experiments involving Rapamycin treatment and social behavior, two-way ANOVA with post hoc analysis was used. Cumulative distributions were analyzed using the Kolmogorov–Smirnov test. All bar graphs represent mean ± SEM.

**Reporting Summary**. Further information on research design is available in the Nature Research Reporting Summary linked to this article.

## Data availability

A reporting summary for this article is available as Supplementary information file. Source data for each main and supplementary figure are provided with this paper.

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

## Acknowledgements
We thank Drs. Elsa Rossignol, Philippe Major (CHU Ste. Justine, Montreal, Canada), and Dr. Fabrice Ango (CNRS, Montpellier, France) for their insightful suggestions and Dr. Guy Doucet (Université de Montréal, Montreal, Canada) for providing reagents for Electron Microscopy. We would like to thank Antônia Samia Fernandes do Nascimento for her technical assistance, Marisol Lavertu-Jolin for helping with quantification of putative synapses in the olfactory bulb, the Comité Institutionnel de Bonne Pratiques Animales en Recherche (CIBPAR), all the personnel of the animal facility of the Research Center of CHU Sainte-Justine (Université de Montreal), Compute Canada and the Plateforme Imagerie Microscopique of the Research Center of CHU Sainte-Justine for their instrumental technical support. This work was supported by the Canadian Institutes of Health Research (A.S and G.DC), Canada Foundation for Innovation (G.DC), Canada Research Chair Program (G.DC), and Natural Sciences and Engineering Research Council of Canada (NSERC). C.A.A. is supported by an NSERC fellowship.

## Author contributions
C.A.A., M.C., A.S., and G.D.C. designed the experiments. C.A.A., M.C., V.J., M.S., and B.C. performed the experiments. C.A.A., M.C., V.J., A.Q., M.S., B.C., A.S., and G.D.C. analyzed data. J.N.C. and M.B. provided critical technical support. C.A.A., M.C., and G.D.C. wrote the manuscript. All authors read and corrected the manuscript.

## Competing interests
The authors declare no competing interests.
