## [Peer Review File · Nature Communications]

Reviewer #1 (Remarks to the Author):

Here the influence of TSC-mTORC1 signaling on the development of PV interneurons in the cerebral cortex is examined and the impact on social interaction is tested. The authors use cell specific Cre recombinase approaches to delete TSC1 from PV interneurons at early developmental time points, including PV-Cre and Tg(Nkx2.1-Cre) mutant mice. The developmental profile of PV interneurons, including synapse and cell number is examined in these models, which include both in vivo and in vitro organotypic approaches. The results demonstrate a premature increase in PV perisomatic synapses at early time points (P18) followed by a decrease at later time points around the 4th week of postnatal development, which also persist to later time points. The results also show a corresponding decrease in social interaction behaviors. In addition, the authors find that administration of rapamycin rescues both the inhibitory synaptic deficit as well as the social interaction deficits in the PV-Cre deletion mutant mice. This work extends previous studies to examine the role of TSC-mTORC1 signaling on development of inhibitory as well as excitatory neurons and the impact of these changes on social interaction. There are several points to address.

1. In Figure 1, the results show that pS6 colocalizes with PV neurons during early postnatal development, and that the number of colocalizations increases from P18-P26. In contrast, there is no increase in NeuN stained cells which are predominantly excitatory neurons. Phospho-S6 is used as a proxy measure of TSC-mTORC1 in these studies, but since mTORC1 and downstream signaling including S6K can be influenced by multiple pathways this is a limitation that should be addressed in the paper.
2. The authors should also show single labeling for NeuN and pS6, as well as higher power images as shown for PV double labeling.
3. In Figure 2 it is difficult to see the appositions of PV with gephyrin above the background gephyrin labeling?
4. In Figure 2M, the distance in open field should be shown across time as there appears to be a trend for increased distance which would likely be more obvious during the early period of the open field. This could significantly influence the other behaviors tested.
5. In addition to PV-Cre to for cell specific deletion of TSC1, tg(Nkx2.1-Cre) mice are also used to allow for better control at earlier time points. However, the NKx2.1 promoter is expressed in both PV and SST interneurons, and deletion in the SST neurons complicates interpretation of the results. This should be addressed.
6. The authors state that the Tg(Nkx2.1-Cre) mice phenocopy the PV-Cre TSC deletion mice, but there are several significant differences. These mice show significant increases in open field distance indicating that they spend much more time moving around the cage which could explain in part the deficits in social interaction. The results also show that the flox/flox mice show a decrease in open arm time, opposite to what was observed in the PV-Cre deletion mice. These points should be addressed.
7. Rapamycin is administered to rescue the synaptic and behavioral deficits observed in the PV-Cre TSC deletion mice. What is the efficacy of rapamycin to block mTORC1 activity when administered ip? This should be tested by analysis of pS6 levels.
8. There was a dissociation between rescue of PV+/Geph+ puncta and social interaction behavior. The authors explain this by lack of rescue of cerebellar effects. It would be important to test this by local knockdown of TSC in specific cortical vs. cerebellar circuits to test this hypothesis and provide more causal data for the observed behavioral deficits.

Reviewer #2 (Remarks to the Author):

In this article by Choudhury et al., the authors investigate the impact of Tsc1, an ASD gene, onto the developmental trajectory of Parvalbumin (PV) expressing interneurons (PV INs), which play critical roles in cortical circuits and whose dysfunction is involved in neurodevelopmental disorders. To this aim, they use an elegant combination of *in vivo* and *ex vivo* experiments to go from single cell inactivation, kinetics of the phenotypes, impact on behavior, molecular mechanism and phenotypic rescue as interventional therapy. First, the authors examine mTOR signaling in PV INs and the increased activity triggered by Tsc1 inactivation selectively in these cells, using PVcre line. They further show that *in vivo* inactivation leads to a hypo-connectivity and social behavior deficits, both hallmarks of Tsc1-associated diseases.

Then they switch to a powerful experimental design set up by the same team, several years ago, to inactivate Tsc1 in single PV cells at different timepoints and find a striking early postnatal hyperconnectivity, that precedes the later observed hypoconnectivity. Remarkably, the hyperconnectivity deficit is rescued at the single cell level by rapamycin treatment, revealing that it relies on mTOR activation. To further test *in vivo*, the authors switch to another cre line model, the Nkx2.1-cre, which drives earlier recombination than the PV line and show in this other system a similar *ex vivo* hyperconnectivity followed by a hypoconnectivity. Taking advantage of this early-recombining model, they showed a transient autophagy dysfunction in these mice, using olfactory bulb samples to bypass the limited amount of proteins. Finally, to determine when Tsc1 and mTOR pathway act in PV cells, the authors deplete Tsc1 late with no impact and perform an *in vivo* rescue by exposing the mice to rapamycin for a week, leading to a full rescue of connectivity in young adults as well as a rescue of the behaviors in heterozygotes.

Collectively this study, by combining a set of experimental approaches carefully quantified, by assessing both heterozygotes and homozygotes in most experiments and by performing an essential timeline of the developmental trajectory will have a major impact in the field. First, it reveals up to the single cell-autonomous level the connectivity changes that can be caused by mTOR overactivation, potentially linked to autophagy. Furthermore, by going to the cell level up to the circuit and behavior, it shows the involvement of mTOR pathway in precise steps of connectivity. Finally, it highlights the need to assess the temporality of the phenotypes to design appropriate therapeutic approaches and reveals a precise timewindow for that.

I thus believe that it will be of great interest for the broad readership of Nature Communications. I nonetheless have a few issues that the authors could address before publication.

Specific point:

- The authors nicely show that the temporal window that matters is the third postnatal week (Figs 1, 3-5,7,8), when PV neurons are hyper connected. However, they do not show *in vivo* this phenotype in the PVcre mutants, which is then used for the *in vivo* rescue. Although this experiment is time consuming and the authors have convincingly used *ex vivo* analyses, I believe that this would add to the strength of the manuscript. In particular, it would be key to decipher whether you need to go through a hyperconnectivity phase to trigger the hypoconnectivity one.

Minor points:

- Throughout the text some specific points could be modified to help out the reader go through the series of experimental models and approaches:

° p4 last paragraph: the authors state “pS6 signalling is increased between the second and fourth postnatal week” but show data in Fig1 at P18 and P26, which are in the third and fourth postnatal week. The authors should adapt their statement to fit the data presented

° Figure 2, the panels o and p could be modified, such as in the other following figures to show the distinct phenotypes with color code (as in figure 8). Because there are several genetic and ex vivo models, it is helpful for the reader to keep consistency across the figures

° p11 the ex vivo biolistics experimental approach could be presented in a more detailed manner in Figure S3. For instance, the EP abbreviation is introduced after (p13) and not necessarily easy to follow for the non-experts.

- All the ex vivo experiments were performed in the “occipital cortex” (p43) whereas the in vivo experiments were imaged in the “somatosensory cortex layers 2/3 and 5/6) (p45). It would be important somewhere in the text to state whether the phenotype is conserved across areas and/or layers, or whether this point does not really matter compared to what is already known in Tsc1/2 models.

Reviewer #3 (Remarks to the Author):

In this manuscript Choudhury et al. propose to investigate the effect of dysfunctional mTORC1 signaling on cortical GABAergic interneurons development and connectivity. For this purpose, they use Tsc1 gene manipulation in PV-expressing neurons and mouse models. The authors show that mutant mice with Tsc1 gene knockout in PV neurons have reduced GABAergic connectivity and altered social behavior at P60. To investigate the cell autonomous effects they perform Tsc1 gene loss in single PV cells of organotypic slices at EP10 and find premature increase in axonal branching at EP18 followed by loss of connectivity at EP34. Interestingly, 1-week rapamycin treatment from EP12 until EP18 rescues both excessive axonal branching and PV innervation supporting a cell-autonomous role of Tsc1/2 in GABAergic neurons. Since connectivity was not lost when performing Tsc1 gene deletion biolistically at EP26 they conclude that excessive pruning at later times is a secondary effect of pre-mature PV synapse formation and altered AMPK-dependent autophagy due to mTORC1 aberrant activity in early post-natal ages. When testing the effect of early postnatal rapamycin treatment in the organotypic cultures they observe complete rescue of PV neurons innervation at EP34 under Tsc1 haploinsufficiency and partial rescue in the cultures from the homozygous Tsc1/ NKX2.1 mice.

This manuscript provides important novel insights on the role of Tsc1/2 complex in inhibitory neurons, uncovers the cell-autonomous morphological and behavioral abnormalities associated with Tsc1 haploinsufficiency in these cells and identifies a sensitive period of intervention to reverse the deficits. The experimental methodology is well designed and properly executed. The use of multiple in vitro and in vivo models provides a robust set-up for the proposed work however, here are specific comments that should be addressed to fully support the conclusions of this manuscript:

1. In Figure 1 the authors show that S6 phosphorylation increases from P18 to P26 in the PV-expressing neurons of the somatosensory cortex and it does not significantly change in the NeuN positive neurons. Given the subsequent finding of altered GABA-ergic neuronal maturation under loss of Tsc1 gene, a prediction would be that at later ages of neuronal development mTORC1 activity should decrease. It would be informative to assess levels of S6 phosphorylation in the somatosensory cortex of older (P60) mice and compared it to the levels of mice in the second and the fourth week.

2. To test whether a phenotypic switch is present in vivo the authors perform single cell Tsc1 gene loss in PV neurons from organotypic cultures of the Tsc1/ NKX2.1 cre mice and confirm a more complex innervation at earlier ages (EP18) followed by loss of connectivity at later times (EP34). Mutant Tsc1/ NKX2.1 cre mice exhibit social deficits, dys-regulated autophagy and increased AMPK activation temporally overlapping with the altered maturation seen in PV neurons of the organotypic cultures of these mice. The authors conclude that altered PV connectivity is a result of dys-regulated AMPK-dep autophagy at the earlier ages.

As in vitro model for these experiments they perform western blots from protein lysates of the olfactory bulb of the Tsc1/ NKX2.1 cre mice and find increased AMPK activity and LC3-II accumulation in GABAergic neurons from younger mice at P14 but not later at P40 (figure 6). To fully support their conclusion the authors should examine: i) whether between P14-P40 the GABAergic neurons of the olfactory bulb undergo the same phenotypic switch observed in the cortex; ii) level of activation of the AMPK-dependent substrate the autophagic kinase ULK1 (p-ULK1 S555) at P14; i) efficacy of early treatment (EP12 to EP18) of the Tsc1/ NKX2.1 cre organotypic cultures with AMPK-specific inhibitors in reversing excessive pruning at EP34.

Figure 6: The authors should also specify which AMPK phosphorylation site was quantified in the western blot in panel B and G. p-AMPK quantification levels should be quantified as the ratio of phosphorylated/ total AMPK levels after protein normalization to GAPDH.

3. When addressing efficacy of rapamycin in vivo, the authors found that mTORC1 inhibition during the third postnatal week rescued PV innervation in both the conditional heterozygous and homozygous PV cre Tsc1 mice at P45. While rapamycin also completely rescued the social behavior deficits in the conditional heterozygous PV cre Tsc1 mice it did not reverse it in the conditional homozygous. The authors conclude that loss of PV cell connectivity in adult mice with Tsc1 gene haploinsufficiency is a result of premature formation of PV cell innervation during a critical period.

To fully support their conclusion the authors should examine whether rapamycin treatment during the early postnatal development reverses the premature axonal complexity in the heterozygous Tsc1/ NKX2.1 cre mice.

Answer to Reviewers' comments.

We are very grateful to the reviewers for their careful assessment of our work and thoughtful suggestions. We have performed new experiments (new Figures 6, 8,9; new Supplemental Figure 8) and revised the paper's discussion to address all the comments. Please find a point by point response below. Changes in the revised manuscript are indicated in red.

Reviewer #1 (Remarks to the Author):

Here the influence of TSC-mTORC1 signaling on the development of PV interneurons in the cerebral cortex is examined and the impact on social interaction is tested. The authors use cell specific Cre recombinase approaches to delete TSC1 from PV interneurons at early developmental time points, including PV-Cre and Tg(Nkx2.1-Cre) mutant mice. The developmental profile of PV interneurons, including synapse and cell number is examined in these models, which include both in vivo and in vitro organotypic approaches. The results demonstrate a premature increase in PV perisomatic synapses at early time points (P18) followed by a decrease at later time points around the 4th week of postnatal development, which also persist to later time points. The results also show a corresponding decrease in social interaction behaviors. In addition, the authors find that administration of rapamycin rescues both the inhibitory synaptic deficit as well as the social interaction deficits in the PV-Cre deletion mutant mice. This work extends previous studies to examine the role of TSC-mTORC1 signaling on development of inhibitory as well as excitatory neurons and the impact of these changes on social interaction. There are several points to address.

1. In Figure 1, the results show that pS6 colocalizes with PV neurons during early postnatal development, and that the number of colocalizations increases from P18-P26. In contrast, there is no increase in NeuN stained cells which are predominantly excitatory neurons. Phospho-S6 is used as a proxy measure of TSC-mTORC1 in these studies, but since mTORC1 and downstream signaling including S6K can be influenced by multiple pathways this is a limitation that should be addressed in the paper.

We agree with the reviewer's comment and we addressed this point in the revised manuscript in the result section when we introduced the first results using pS6 as a marker (Supplemental Figure 1).

2. The authors should also show single labeling for NeuN and pS6, as well as higher power images as shown for PV double labeling.

Done as suggested, please see revised Figure 1.

3. In Figure 2 it is difficult to see the appositions of PV with gephyrin above the background gephyrin labeling?

We thank the reviewer for pointing out this issue. We chose new images and changed the color code to be consistent with Figure 10.

4. In Figure 2M, the distance in open field should be shown across time as there appears to be a trend for increased distance which would likely be more obvious during the early period of the open field. This could significantly influence the other behaviors tested.

We quantified the distanced covered by each mouse in the open field in the first 5 minutes (the test lasts 10 minutes), and plotted Mean±SEM in the graph below. We did not detect any significant difference between the genotypes (N of mice, as indicated in Figure 2M; one-way Anova, $p=0.4705$).

5. In addition to PV-Cre to for cell specific deletion of TSC1, tg(Nkx2.1-Cre) mice are also used to allow for better control at earlier time points. However, the NKx2.1 promoter is expressed in both PV and SST interneurons, and deletion in the SST neurons complicates interpretation of the results. This should be addressed.

We agree with the reviewer's comment and we addressed this issue in the discussion in the revised manuscript. Briefly, a recent study reported that *Tsc1* deletion specifically in somatostatin (SST)-expressing GABAergic interneurons (*SST-Cre;Tsc1^{lox/+}* mice) lead to ectopic expression of PV and to altered firing properties in a small percentage of cortical SST neurons in conditional homozygous and heterozygous mice but to overall reduced synaptic output only in the conditional homozygous mutants (Malik et al, 2019, reference 12). Our data (including new figures 8 and 9) demonstrate that both *Nxk2.1Cre;Tsc1^{lox/+}* and *PVCre;Tsc1^{lox/+}* conditional heterozygous mice show an altered developmental time course of putative synapses formed by PV cells, with a premature hyper-connectivity in pre-adolescent mice followed by a significant hypo-connectivity in adults. In addition, both our conditional heterozygous mice showed altered social behaviors. Over all, these data suggest that the formation and refinement of PV cell synaptic connectivity is more sensitive to *Tsc1* haploinsufficiency than SST neurons even if both these GABAergic interneuron populations originate from the medial ganglionic eminence.

6. The authors state that the Tg(Nkx2.1-Cre) mice phenocopy the PV-Cre TSC deletion mice, but there are several significant differences. These mice show significant increases in open field distance indicating that they spend much more time moving around the cage

which could explain in part the deficits in social interaction. The results also show that the flox/flox mice show a decrease in open arm time, opposite to what was observed in the PV-Cre deletion mice. These points should be addressed.

We described and discussed these difference between the conditional homozygous lines, *Tg(Nkx2.1-Cre);Tsc1^{lox/lox}* and *Tg(PV-Cre);Tsc1^{lox/lox}* mice, in the results session of the revised manuscript. Briefly, we speculate that differences in the time spent in the open arms of the elevated plus maze or in the distance covered on the open field may due to the different GABAergic circuits affected in the two mouse lines (Nkx2.1 is expressed by somatostatin-positive neurons as well) or/and by the different timing of *Tsc1* deletion (embryonal vs postnatal). We would like to point out that both conditional heterozygous lines (*Tg(Nkx2.1-Cre);Tsc1^{lox/+}* and *Tg(PV-Cre);Tsc1^{lox/+}*) do not show any significant differences compared to their wild-type littermates neither in the open field nor in the elevated plus maze tests (Fig.2M, N and Suppl Figure 7), however they both show similarly impaired social behavior. We think these observations are relevant because in patients, Tuberous Sclerosis is a dominant disorder caused by mutations in just one *Tsc1* (or *Tsc2*) allele.

7. Rapamycin is administered to rescue the synaptic and behavioral deficits observed in the PV-Cre TSC deletion mice. What is the efficacy of rapamycin to block mTORC1 activity when administered ip? This should be tested by analysis of pS6 levels.

To answer this question, we would need to analyse pS6 expression levels immediately after the end of the 7 days-long treatment in *PVCre;Tsc1^{lox}* mice *in vivo*. Since in our experience, Cre expression does not reach plateau until the 3rd postnatal week in PVCre mice, we decided to introduce a reported allele, the RCE allele, which has a stop codon flanked by two lox sites before the sequence coding for eGFP. eGFP expression allowed us to identify the PV cells where *Tsc1* recombination had likely already occurred (new figures 8, 9). Consistently to what we observed in mice with embryonic deletion of *Tsc1* in MGE-derived GABAergic cells, postnatal *Tsc1* haploinsufficiency in PV cells caused a premature increased of PV+ perisomatic synapse density (Fig. 8G). We then treated *PV-Cre;RCE;Tsc1^{lox/+}* and their control littermates (*PV-Cre;RCE;Tsc^{+/+}*) daily with either rapamycin or vehicle from P14 to P21 and analyzed PV cell perisomatic synaptic density at the end of the treatment at P22 (Fig.9A). We found that this treatment was sufficient to rescue the premature PV cell hyperconnectivity at P22 (Fig.9 N, O).

While rapamycin was clearly effective in rescuing hyperconnectivity in *PV-Cre;RCE;Tsc1^{lox/+}* (cHet) mice, to investigate the effect of rapamycin on pS6 levels following IP injection in our model, we would need to analyse pS6 levels after the last injection at P22 in *PV-Cre;RCE;Tsc1^{lox/lox}* (cKO) mice because at this age pS6 is clearly and significantly increased in PV cell somata only in the cKO (Supplemental Figure 1C). Due to time constrains, we generated *PV-Cre;RCE;Tsc1^{+/+}* (ctrl) and *PV-Cre;RCE;Tsc1^{lox/+}* (cHet) mice but not cKO mice. However, the chosen dose range and injection protocol (daily I.P. injections) has been used in several previous studies, which

reported that this treatment significantly reduced pS6 expression levels in transgenic mice carrying *Tsc1* deletions (see for example Suppl Figure 6 in Tsai et al, 2018, reference 45).

8. There was a dissociation between rescue of PV+/Gep+ puncta and social interaction behavior. The authors explain this by lack of rescue of cerebellar effects. It would be important to test this by local knockdown of TSC in specific cortical vs. cerebellar circuits to test this hypothesis and provide more causal data for the observed behavioral deficits.

We thank the reviewer for raising this point. Actually, one of the reasons we generated two different mouse lines to target *Tsc1* haploinsufficiency to PV cells was to better understand the functional consequences of these manipulations on mouse behavior. In fact, no transgenic mouse line can target gene manipulation exclusively to cortical PV cells, since PV is expressed in different brain regions, including the cerebellum. *Nkx2.1*, on the other hand, is expressed by both SST and PV cortical neurons, but not in the cerebellum, a pattern we verified by using a GFP reporter mice.

Our data suggest that *Tsc1* haploinsufficiency in cortical GABAergic cells is sufficient to cause social behavior deficits because *Nkx2.1;Tsc1^{lox/+}* and *Nkx2.1;Tsc1^{lox/lox}* mice do not show preference for a mouse vs an object or for a novel mouse vs a familiar one in the 3 chamber test, while their wild-type littermates do (Suppl Figure 7). In these mutant mice, Cre is not expressed in any cerebellar cell populations. We cannot exclude that *Tsc1* deletion in cortical SST neurons may contribute to the social behavioral phenotypes, however we think this is unlikely because, as mentioned above, only conditional homozygous, but not heterozygous, *SSTCre;Tsc1* mutant mice showed overall reduced inhibitory synaptic output (Malik et al, 2019, reference 12).

On other hand, social behavior deficits can arise from alterations in different circuits and brain regions, including PV cell circuits in prefrontal cortex (see for example, Cao et al, Neuron 2018, PMID29503190; Selimbeyoglu et al, Sci Transl Med, 2017, PMID 28768803) and Purkinje cells in the cerebellum (Tsai et al, 2012, reference 38). In fact, Sahin's group published a seminal paper showing that *Tsc1* deletion exclusively in Purkinje cells (PCs) was sufficient to cause social behavior deficits (Tsai et al, 2012, reference 38). PV is expressed by a subset of adult PCs, therefore the alterations we found in PC morphology in adult *PVCre;Tsc1^{lox/lox}* mice are consistent to those reported by Tsai et al (2012) in PC-specific *Tsc1* cKO mice. The same group published a more recent study investigating the efficiency of rapamycin treatment during different developmental stages in rescuing social behavior and PC-physiology deficits in PC-specific *Tsc1* cKO mice (*L7Cre;Tsc1^{lox/lox}* mice; Tsai et al, 2018). A four weeks-long rapamycin treatment initiated at either at P7 or 6 weeks after birth rescued the deficits of both social behavior and intrinsic firing properties and excitability of PCs in the mutant mouse. It therefore likely that a 7 days treatment from P14-P21 might be too short to rescue PC morphology and function in *PVCre;Tsc1^{lox/lox}* (cKO) mice. Conversely, we did not find any major morphological differences in PCs in *PVCre;Tsc1^{lox/+}* (cHet), thus leaving cortical alterations of PV cells as a likely major contributor of the observed social behavioral deficits.

Reviewer #2 (Remarks to the Author):

In this article by Choudhury et al., the authors investigate the impact of Tsc1, an ASD gene, onto the developmental trajectory of Parvalbumin (PV) expressing interneurons (PV INs), which play critical roles in cortical circuits and whose dysfunction is involved in neurodevelopmental disorders. To this aim, they use an elegant combination of *in vivo* and *ex vivo* experiments to go from single cell inactivation, kinetics of the phenotypes, impact on behavior, molecular mechanism and phenotypic rescue as interventional therapy. First, the authors examine mTOR signaling in PV INs and the increased activity triggered by Tsc1 inactivation selectively in these cells, using PV^{cre} line. They further show that *in vivo* inactivation leads to a hypo-connectivity and social behavior deficits, both hallmarks of Tsc1-associated diseases.

Then they switch to a powerful experimental design set up by the same team, several years ago, to inactivate Tsc1 in single PV cells at different timepoints and find a striking early postnatal hyperconnectivity, that precedes the later observed hypoconnectivity. Remarkably, the hyperconnectivity deficit is rescued at the single cell level by rapamycin treatment, revealing that it relies on mTOR activation. To further test *in vivo*, the authors switch to another cre line model, the Nkx2.1-cre, which drives earlier recombination than the PV line and show in this other system a similar *ex vivo* hyperconnectivity followed by a hyperconnectivity deficit. Taking advantage of this early-recombining model, they showed a transient autophagy dysfunction in these mice, using olfactory bulb samples to bypass the limited amount of proteins. Finally, to determine when Tsc1 and mTOR pathway act in PV cells, the authors deplete Tsc1 late with no impact and perform an *in vivo* rescue by exposing the mice to rapamycin for a week, leading to a full rescue of connectivity in young adults as well as a rescue of the behaviors in heterozygotes.

Collectively this study, by combining a set of experimental approaches carefully quantified, by assessing both heterozygotes and homozygotes in most experiments and by performing an essential timeline of the developmental trajectory will have a major impact in the field. First, it reveals up to the single cell-autonomous level the connectivity changes that can be caused by mTOR overactivation, potentially linked to autophagy. Furthermore, by going to the cell level up to the circuit and behavior, it shows the involvement of mTOR pathway in precise steps of connectivity. Finally, it highlights the need to assess the temporality of the phenotypes to design appropriate therapeutic approaches and reveals a precise timewindow for that.

I thus believe that it will be of great interest for the broad readership of Nature Communications. I nonetheless have a few issues that the authors could address before publication.

Specific point:

- The authors nicely show that the temporal window that matters is the third postnatal week (Figs 1, 3-5,7,8), when PV neurons are hyper connected. However, they do not show in vivo this phenotype in the PVcre mutants, which is then used for the in vivo rescue. Although this experiment is time consuming and the authors have convincingly used ex vivo analyses, I believe that this would add to the strength of the manuscript. In particular, it would be key to decipher whether you need to go through a hyper connectivity phase to trigger the hypo connectivity one.

We wholeheartedly agreed with the reviewer on the importance of this experiment, even if time consuming. Since in our experience, Cre expression does not reach plateau until the 3rd postnatal week in this PVCre line, we generated *PV-Cre;RCE;Tsc1^{+/+}* and *PV-Cre;RCE;Tsc1^{lox/+}* mice. As explained above, the RCE allele has a stop codon flanked by two lox sites before the sequence coding for eGFP. eGFP expression allowed us to identify the PV cells where Tsc1 recombination had likely already occurred. (new figure 8). Consistently to what we observed in mice with embryonic deletion of *Tsc1* in MGE-derived GABAergic cells, postnatal *Tsc1* haploinsufficiency in PV cells caused a premature increased of PV+ perisomatic synapse density (Fig.8G). We then treated *PV-Cre;RCE;Tsc1^{lox/+}* and their control littermates (*PV-Cre;RCE;Tsc1^{+/+}*) daily with either rapamycin or vehicle from P14 to P21 and analyzed PV cell perisomatic synaptic density at the end of the treatment (new figure 9). We found that this treatment was sufficient to rescue the premature PV cell hyper-connectivity (Fig.9 N, O). Overall, these new data support the hypothesis that the hyper-connectivity phase during a sensitive developmental time window is critical to trigger excessive pruning leading to hypo-connectivity in adult mice.

Minor points:

- Throughout the text some specific points could be modified to help out the reader go through the series of experimental models and approaches:

° p4 last paragraph: the authors state “pS6 signalling is increased between the second and fourth postnatal week” but show data in Fig1 at P18 and P26, which are in the third and fourth postnatal week. The authors should adapt their statement to fit the data presented

Thank you for noticing. We made the required changes.

° Figure 2, the panels o and p could be modified, such as in the other following figures to show the distinct phenotypes with color code (as in figure 8). Because there are several genetic and ex vivo models, it is helpful for the reader to keep consistency across the figures

Done as suggested.

° p11 the ex vivo biolistics experimental approach could be presented in a more detailed manner in Figure S3. For instance, the EP abbreviation is introduced after (p13) and not necessarily easy to follow for the non-experts.

Done as suggested.

- All the ex vivo experiments were performed in the “occipital cortex” (p43) whereas the in vivo experiments were imaged in the “somatosensory cortex layers 2/3 and 5/6) (p45). It would be important somewhere in the text to state whether the phenotype is conserved across areas and/or layers, or whether this point does not really matter compared to what is already known in Tsc1/2 models.

Actually, the statement in the material and methods section was imprecise, we apologize. When we prepare organotypic cultures, we use all the brain slices obtained from slicing the brain starting at the occipital cortex until the end of the somatosensory cortex. In the last 15 years, we have imaged and analysed dozens of PV+ cells labeled with biolistic transfection and we have not observed any major differences in their perisomatic connectivity with respect to the cortical region where they are located. We corrected the statement in the methods.

Reviewer #3 (Remarks to the Author):

In this manuscript Choudhury et al. propose to investigate the effect of dysfunctional mTORC1 signaling on cortical GABAergic interneurons development and connectivity. For this purpose, they use Tsc1 gene manipulation in PV-expressing neurons and mouse models. The authors show that mutant mice with Tsc1 gene knockout in PV neurons have reduced GABAergic connectivity and altered social behavior at P60. To investigate the cell autonomous effects they perform Tsc1 gene loss in single PV cells of organotypic slices at EP10 and find premature increase in axonal branching at EP18 followed by loss of connectivity at EP34. Interestingly, 1-week rapamycin treatment from EP12 until EP18 rescues both excessive axonal branching and PV innervation supporting a cell-autonomous role of Tsc1/2 in GABAergic neurons. Since connectivity was not lost when performing Tsc1 gene deletion biolistically at EP26 they conclude that excessive pruning at later times is a secondary effect of pre-mature PV synapse formation and altered AMPK-dependent autophagy due to mTORC1 aberrant activity in early post-natal ages. When testing the effect of early postnatal rapamycin treatment in the organotypic cultures they observe complete rescue of PV neurons innervation at EP34 under Tsc1 haploinsufficiency and partial rescue in the cultures from the homozygous Tsc1/ NKX2.1 mice.

This manuscript provides important novel insights on the role of Tsc1/2 complex in

inhibitory neurons, uncovers the cell-autonomous morphological and behavioral abnormalities associated with *Tsc1* haploinsufficiency in these cells and identifies a sensitive period of intervention to reverse the deficits. The experimental methodology is well designed and properly executed. The use of multiple *in vitro* and *in vivo* models provides a robust set-up for the proposed work however, here are specific comments that should be addressed to fully support the conclusions of this manuscript:

1. In Figure 1 the authors show that S6 phosphorylation increases from P18 to P26 in the PV-expressing neurons of the somatosensory cortex and it does not significantly change in the NeuN positive neurons. Given the subsequent finding of altered GABA-ergic neuronal maturation under loss of *Tsc1* gene, a prediction would be that at later ages of neuronal development mTORC1 activity should decrease. It would be informative to assess levels of S6 phosphorylation in the somatosensory cortex of older (P60) mice and compared it to the levels of mice in the second and the fourth week.

We thank the reviewer for raising this point. We actually quantified the percentage of PV cells expressing detectable levels of pS6 at different developmental time point and in adult mice when we first started the project. After the fourth postnatal week, the proportion of PV cells expressing pS6 remained stable (P26: $75 \pm 7\%$, P35: $70 \pm 7\%$; P150: $78 \pm 3\%$, one-way Anova, $p > 0.1$; $n = 3$ for each age group). We report this quantification in the result section of the revised manuscript. One possible explanation is that increased mTORC1 signaling during the third and fourth postnatal weeks might play a critical role in the proliferation and refinement of PV cell circuits and the maturation of their physiological properties, which are known to occur during this period. Indeed, our data suggest that counteracting the dysregulation of mTORC1 in PV cells during this sensitive developmental window is sufficient to rescue their connectivity and social behavior. We cannot exclude that mTORC1 in adult PV cells might still play a subtle role in their metabolic state or plasticity.

2. To test whether a phenotypic switch is present *in vivo* the authors perform single cell *Tsc1* gene loss in PV neurons from organotypic cultures of the *Tsc1/ NKX2.1* cre mice and confirm a more complex innervation at earlier ages (EP18) followed by loss of connectivity at later times (EP34). Mutant *Tsc1/ NKX2.1* cre mice exhibit social deficits, dys-regulated autophagy and increased AMPK activation temporally overlapping with the altered maturation seen in PV neurons of the organotypic cultures of these mice. The authors conclude that altered PV connectivity is a result of dys-regulated AMPK-dep autophagy at the earlier ages. As *in vitro* model for these experiments they perform western blots from protein lysates of the olfactory bulb of the *Tsc1/ NKX2.1* cre mice and find increased AMPK activity and LC3-II accumulation in GABAergic neurons from younger mice at P14 but not later at P40 (figure 6). To fully support their conclusion the authors should examine: i) whether between P14-P40 the GABAergic neurons of the olfactory bulb undergo the same phenotypic switch observed in the cortex; ii) level of activation of the AMPK-dependent substrate the autophagic kinase ULK1 (p-ULK1 S555) at P14; i) efficacy of early treatment (EP12 to EP18) of the *Tsc1/ NKX2.1* cre organotypic cultures with AMPK-specific inhibitors in reversing excessive pruning at

EP34.

- i) To investigate whether GABAergic synapse density in the olfactory bulb were affected by *Tsc1* deletion, we imaged the external plexiform layer (EPL), since this layer is highly enriched in PV cells (Takeshi 2014; PMID 25084319). In the EPL, PV-positive cells are typically axonless and their multipolar dendrites form dendro-dendritic synapses, which can be identified by immunolabeling for the vesicular GABA transporter (vGAT) and gephyrin (Matsuno et al., 2017, PMID 27864926). We found a significant reduction in adult mutant mice (new supplemental figure 8), similarly to what we observed for cortical PV+Gephyrin+ density (Fig.2A-D).
- ii) We analyzed the level of activation of p-ULK1 (S555) in the same P14 samples we used for pAMPK analysis by western blot. We found that pAMPK/total AMPK was significantly increased ($p=0.003$), while pULK/total ULK showed a trend towards an increase ($p=0.065$) (new figure figure 6)
- iii) We performed preliminary experiments as suggested by the reviewer by applying the AMPK inhibitor Compound C for 6 days from EP12 to EP18. We tried two different doses, in the range used in published studies and by A. Saghatelian, one of the author of this study. Unfortunately, we observed non-specific effects on neuron morphology and on the overall health state of the organotypic cultures. Compound C has been used in many different studies *in vivo* and in acute brain slices but mostly for short term application (ranging from 30 minutes to 2-3 hrs). It is possible that inhibiting AMPK for several days negatively affects the metabolic states of brain slice tissue. Few years ago, we actually cloned both AMPK constitute active and dominant negative forms under the promoter that allows selective expression in PV cells in organotypic cultures by biolistic transfection. Preliminary experiments showed unspecific effects of both constructs on PV cell morphology. Overall, transfected PV cells looked quite unhealthy. It is likely that balanced AMPK activity, possibly at specific subcellular locations, is needed for its physiological function. Overall, our observation that pAMPK is increased following *Tsc1* deletion in GABAergic neurons derived from the medial ganglionic eminence is consistent with a recent study that showed that, in neurons, *Tsc2/1* loss altered the process of autophagy by AMPK-dependent mechanisms (reference 42). However, further studies will be needed to address the specific effects of AMPK hyper-phosphorylation in *Tsc1* haploinsufficient, cortical PV cells. We now discuss the limitation of these findings in the discussion of the revised manuscript.

Figure 6: The authors should also specify which AMPK phosphorylation site was quantified in the western blot in panel B and G. p-AMPK quantification levels should be quantified as the ratio of phosphorylated/ total AMPK levels after protein normalization to GAPDH.

We thank the reviewer for this comment. We used an antibody against pAMPK T172, as reported in the methods section of the revised manuscript. As suggested by the reviewer, we quantified and reported pAMPK/total AMPK.

3. When addressing efficacy of rapamycin in vivo, the authors found that mTORC1 inhibition during the third postnatal week rescued PV innervation in both the conditional heterozygous and homozygous PV cre *Tsc1* mice at P45. While rapamycin also completely rescued the social behavior deficits in the conditional heterozygous PV cre *Tsc1* mice it did not reverse it in the conditional homozygous. The authors conclude that loss of PV cell connectivity in adult mice with *Tsc1* gene haploinsufficiency is a result of premature formation of PV cell innervation during a critical period.

To fully support their conclusion the authors should examine whether rapamycin treatment during the early postnatal development reverses the premature axonal complexity in the heterozygous *Tsc1*/NKX2.1 cre mice.

We completely agree with the reviewer on the importance of this experiment. We decided to perform this experiment in *PVCre;Tsc1^{lox/+}* mice to better compare the results with the analysis of the rescue experiments at >P45. As explained above, we generated *PV-Cre;RCE;Tsc1^{+/+}* and *PV-Cre;RCE;Tsc1^{lox/+}* mice so we could identify the PV cells where *Tsc1* recombination had likely already occurred. Consistently to what we observed in mice with embryonic deletion of *Tsc1* in MGE-derived GABAergic cells, postnatal *Tsc1* haploinsufficiency in PV cells caused a premature increased of PV+ perisomatic synapse density (new figure 8). We then treated *PV-Cre;RCE;Tsc1^{lox/+}* and their control littermates (*PV-Cre;RCE;Tsc1^{+/+}*) daily with either rapamycin or vehicle from P14 to P21 and analyzed PV cell perisomatic synaptic density at the end of the treatment (new figure 9). We found that this treatment was sufficient to rescue the premature PV cell hyperconnectivity (Fig.9 N, O). Overall, these data support the hypothesis that the hyperconnectivity phase during a sensitive developmental time window is critical to trigger excessive pruning leading to hypo-connectivity in adult mice.

Reviewer #2 (Remarks to the Author):

In this revised version of the manuscript, the authors have addressed the main points raised by the reviewers.

In particular reviewers 1 and 2 had pointed out the importance to compare in more depth the two genetic models of Tsc1 inactivation, using either the Nkx2.1-cre or the PV-cre to address the selectivity of the phenotype (PV or SST interneurons, as well as cortex versus cerebellum). The authors, by adding several key experiments in new Figures 8 and 9, show that the hyper connectivity is observed in both models and that the postnatal rapamycin treatment efficiently blocks this phenotype. They also provide key additional experiments on the models regarding behavioral outputs as well as rapamycin treatment administration. Furthermore, authors have put in a broader perspective, as requested, the analyses of pS6 staining as readout of the mTOR pathway. Finally, the authors have remodelled the figures and text to address all the minor comments or explanations requested.

In its present form, this study puts forward extremely convincing and exciting results on the roles of mTOR pathway in the maturation of PV interneurons innervation, with major insights into potential novel therapeutical approaches using rapamycin.

This study should be of broad interest for the readership of Nature Communications and I fully support its publication.

Reviewer #3 (Remarks to the Author):

The authors have fully addressed the comments of Reviewer 1 and Reviewer 3. The manuscript is now acceptable for publication.